# MotionDDM: Motion Generation and Understanding via Discrete Diffusion Model

## Abstract

We present MotionDDM, a diffusion-LLM framework for bidirectional text-motion understanding and generation. Unlike GPT-style autoregressive approaches that tokenize motion and decode sequentially, MotionDDM performs multi-step parallel denoising, unifying Text-to-Motion (T2M), Motion-to-Text (M2T), and text-free Motion-to-Motion (M2M) within a single model. This decoding paradigm naturally enables a quality-latency trade-off at inference. On HumanML3D, our method achieves competitive T2M/M2T results against strong baselines. Beside T2M/M2T, we further demonstrate motion completion, prediction, and interpolation under both text-conditioned and text-free settings. We also incorporate Residual VQ (RVQ) as the motion tokenizer to improve quantization fidelity, and adopt GRPO within the framework to enhance alignment and controllability. To the best of our knowledge, this is the first work to bring diffusion-LLMs to bidirectional text-motion modeling.

## 1 Introduction

Bidirectional text–motion understanding and generation is increasingly important for modern motion systems Guo et al. (2022a). Beyond txt2motion (T2M) and motion2txt (M2T), many practical tasks—such as caption correction, motion continuation, and motion in-betweening—require a model to both interpret motion and generate it. Treating T2M and M2T as separate problems leads to inconsistent representations and duplicated engineering, whereas a unified framework provides shared cross-modal embeddings and consistent behavior across tasks. However, achieving high-quality generation, low-latency decoding, editability, and multi-task unification within a single model remains challenging.

Most prevailing approaches follow the *autoregressive (AR)* paradigm: continuous motions are discretized into "tokens" and decoded sequentially in a GPT-style manner to accomplish *Text-to-Motion (T2M)* and *Motion-to-Text (M2T)* Jiang et al. (2024); Wang et al. (2024); Guo et al. (2022b). While this line of work has advanced unified interfaces and transferability, it faces inherent limitations: (i) token/frame-by-token decoding makes end-to-end latency scale with sequence length; (ii) the forward-only nature of AR decoding hinders global, multi-step correction, which is important for editing, completion, and interpolation; and (iii) unifying text-conditioned and text-free variants of completion, prediction, and interpolation typically requires additional engineering branches and specialized training tricks. Recent module decoupling and cross-modal attention designs alleviate the tug-of-war between language and motion to some extent Zhu et al. (2025), but decoding and editability remain fundamental bottlenecks.

We present **MotionDDM**, a *diffusion-LLM* framework for *bidirectional* text–motion modeling. The key idea is to treat both text and motion as noisy sequences and perform $K$-step parallel denoising that progressively converges under global context. Unlike AR decoding, diffusion-style parallel denoising naturally supports block-level parallelism and multi-step self-correction, enabling a single model to unify T2M, M2T, and text-free *Motion-to-Motion (M2M)*. Crucially, by adjusting the number of denoising steps at inference, the framework provides a tunable *quality–latency* trade-off, allowing practitioners to select Pareto-optimal operating points for different applications Li et al. (2022); Nie et al. (2025). To improve motion representation fidelity, we adopt *Residual Vector Quantization (RVQ)* Zeghidour et al. (2021) as the motion tokenizer, achieving lower quantization

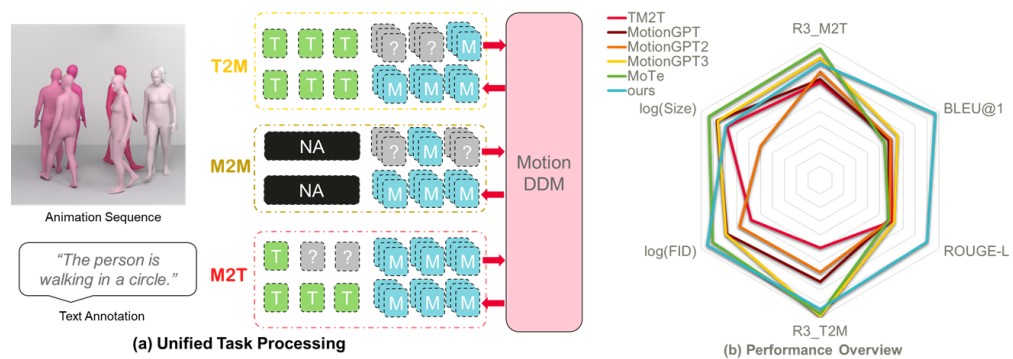

Figure 1: Overview of MotionDDM. Our method achieves a better balance on T2M/M2T tasks.

error at comparable bitrates; moreover, we further incorporate *GRPO* within the framework Shao et al. (2024) to enhance alignment and controllability.

On *HumanML3D* Guo et al. (2022a), our approach attains competitive results against strong baselines for both T2M and M2T, and empirically validates the quality–latency tunability of diffusion decoding. We also prototype text-conditioned and text-free motion completion, prediction, and interpolation, showcasing the unification and extensibility of **MotionDDM**.

Our contributions are threefold:

1. **Paradigm & tunable decoding.** We introduce diffusion-LLMs to *bidirectional* text–motion (T2M↔M2T) and propose a *multi-step parallel denoising* decoder under a unified framework. This paradigm naturally enables a quality–latency trade-off, which we validate with Pareto curves from step-count sweeps.

2. **Unified capability & natural extensibility.** Beyond standard T2M/M2T, our proposed framework can inherently support *M2M* completion, prediction, and interpolation under both text-conditioned and text-free settings on the T2M side, evidencing the paradigm's natural extensibility.

3. **Practical effectiveness & scalability.** We employ *RVQ* as the motion tokenizer to reduce quantization error and improve downstream quality, and integrate *GRPO* to enhance alignment and controllability; In HumanML3D, we achieve competitive results on both T2M and M2T tasks.

## 2 RELATED WORKS

### 2.1 HUMAN MOTION GENERATION

Human motion generation has been a long-standing research problem at the intersection of computer vision, computer graphics, and machine learning. Early works relied on motion graphs and statistical models (Rose et al., 1998; Mukai & Kuriyama, 2005), which enabled interpolation of motion clips but lacked semantic controllability. With the development of deep learning, generative models are introduced into this field. Generative adversarial networks (GANs) (Harvey et al., 2020; Barsoum et al., 2018), variational autoencoders (VAEs) (Aliakbarian et al., 2020; Petrovich et al., 2021), and diffusion-based approaches (Tevet et al., 2022; Zhang et al., 2024a) further advanced the field, improving the diversity and realism of synthesized sequences. Later works also focus on higher motion controllability (Zhang et al., 2024c; Karunratanakul et al., 2023), motion generation under multimodal control signal (Gong et al., 2023; Zhang et al., 2024b).

### 2.2 BI-DIRECTIONAL MOTION-TEXT GENERATION

Bridging motion and natural language has attracted increasing interest for applications such as video captioning, embodied AI, and human-robot interaction. Text-to-motion (T2M) models (Guo et al., 2022a; Petrovich et al., 2022; Zhang et al., 2024a) synthesize 3D motion sequences aligned with

textual prompts. A notable effort toward unification is **MotionGPT** (Jiang et al., 2024), which quantizes human motion into discrete tokens and treats them as part of a shared vocabulary with language, enabling both T2M and M2T through a large language model backbone. While MotionGPT demonstrates strong bi-directional performance, its autoregressive generation can be inefficient for long motion sequences and may struggle to capture structural refinements. **MotionGPT-2** (Wang et al., 2024) extends this line by integrating multimodal controls such as single-frame poses into the same framework. Other bi-directional approaches, such as TM2T (Guo et al., 2022b), typically train separate models for each direction, lacking a unified architecture. Our work differs by combining the bi-directional modeling capacity of MotionGPT with the efficiency of masked refinement.

### 2.3 Diffusion Language Models (dLLMs)

While autoregressive (AR) large language models (LLMs) such as GPT-style transformers (Radford et al., 2018; Brown et al., 2020) dominate current research and applications, an alternative line of work explores *bidirectional denoising-based models*. This family, often referred to as **diffusion language models (dLLMs)**, extends the principles of discrete diffusion (Austin et al., 2021; Nie et al., 2025) and masked language modeling (Devlin et al., 2019) to sequential text generation.

Instead of predicting tokens strictly left-to-right, dLLMs apply a diffusion-based noise schedule that progressively corrupts token sequences, and then learn to iteratively denoise them. Representative works include **Diffusion-LM** (Li et al., 2022), which formulates text generation as discrete denoising diffusion, and **MaskGIT** (Chang et al., 2022), which introduces iterative masked prediction for efficient parallel decoding. Recent surveys (Yu et al., 2025) highlight that this paradigm achieves competitive quality compared to AR models, while offering controllable sampling, parallel decoding, and a natural quality–latency trade-off.

Building on these insights, our proposed **MotionDDM** adapts the dLLM paradigm to motion–language modeling, integrating residual vector quantization for motion tokenization with diffusion-style denoising to support bidirectional generation across text and motion. For the language backbone, we adopt the BERT family, whose bidirectional masked language modeling naturally fits our denoising framework and is well aligned with the size of available motion–language datasets.

## 3 Methods

### 3.1 Overview

We propose **MotionDDM**, a unified framework for *bidirectional text-motion generation*, covering text-to-motion (T2M), motion-to-text (M2T), and text-free motion tasks such as completion, prediction, and interpolation. MotionDDM formulates both text and motion as discrete token sequences, applies a diffusion-style corruption process, and trains the model to iteratively denoise them, enabling multi-step parallel decoding that balances generation quality and latency.

Our design combines three components: (i) a **Residual Vector Quantizer (RVQ)** for high-fidelity motion tokens; (ii) a **BERT-based masked language model backbone** with a motion encoder-decoder, which fuses motion and text embeddings in a shared denoising framework; and (iii) an optional **GRPO-based reinforcement objective** for improved cross-modal alignment. This allows MotionDDM to achieve flexible and high-quality motion-language modeling.

### 3.2 MotionDDM

MotionDDM is a unified framework for bidirectional text-motion generation, inspired by the recent success of *discrete diffusion language models (dLLMs)*. Instead of sequentially autoregressing tokens, dLLMs apply random masking and iterative denoising, which naturally supports **parallel inference**. This allows the model to refine corrupted sequences in multiple steps, dynamically revise low-confidence predictions, and leverage bidirectional attention for stronger contextual reasoning.

**Why Motion Fits Masked Modeling.** Motion data is inherently spatiotemporal: each frame depends on local kinematics while remaining globally constrained by long-range dynamics. Masked denoising is particularly suitable here, as it allows the model to recover missing trajectories us-

ing surrounding context, while iterative refinement prevents error accumulation over long horizons. This makes motion a natural candidate for masked generation, analogous to text but with temporal continuity as an additional inductive bias.

**Multi-task Scheduling.** To unify multiple objectives, MotionDDM employs a **multi-task scheduling** mechanism. During training, each sample within a batch is randomly assigned to one of three tasks: text-to-motion (T2M), motion-to-text (M2T), or motion-to-motion (M2M). The proportion of tasks is controlled by a tunable hyperparameter, enabling fine-grained adjustment of cross-modal versus unimodal supervision. The defination of these three tasks are listed as below:

- **Text-to-Motion (T2M):** The model learns to recover corrupted motion sequences from an unmasked text prompt by reconstructing masked motion tokens. All text tokens are reserved while motion tokens are randomly masked out.
- **Motion-to-Text (M2T):** The model is supposed to translate movement into natural language descriptions, effectively performing motion captioning. All motion tokens are reserved while text tokens are randomly masked out.
- **Motion-to-Motion (M2M):** In this setting, the model focuses solely on motion by recovering masked motion tokens, ignoring any text input. This self-supervised objective enhances the model's capabilities for motion prediction, completion, and interpolation.

**Masking Schedule and Training Loss** We train MotionDDM with a multi-task masked denoising objective. For each sequence $y$ and its textual description $x$, a set of positions $\mathcal{M}$ is masked, and the model predicts the masked tokens from corrupted $\tilde{y}$. The overall loss is masked cross-entropy:

$$\mathcal{L}_{\text{task}} = \mathbb{E}_{(x,y)\sim\mathcal{D}}\Big[ - \sum_{t\in\mathcal{M}} \log p_\theta(y_t \mid \tilde{y}, x)\Big]. \tag{1}$$

We apply a **linear masking schedule**, where the masking probability increases linearly with a random scalar $u \sim \mathcal{U}(0,1)$:

$$p_{\text{mask}} = \lambda \cdot u, \tag{2}$$

with $\lambda \in [0,1]$ controlling the maximum masking ratio. This ensures that short sequences are partially corrupted while longer sequences may be heavily masked, yielding more robust learning. Linear scheduling avoids overly aggressive masking at early steps and provides smoother curriculum than random or cosine schedules.

**Confidence-Guided Progressive Inference.** We adopt a confidence-guided progressive inference strategy, inspired by non-autoregressive masked decoding. Given a partially masked sequence $\tilde{y}$ (motion or text tokens), the model refines it over $S$ denoising steps. At each step $s$, the model predicts a distribution $p_\theta(y_t \mid \tilde{y}, x)$ for all currently masked positions $t$, and estimates token-wise confidence by the maximum probability:

$$c_t = \max_v p_\theta(y_t = v \mid \tilde{y}, x).$$

A subset of masked positions is then committed according to a predefined schedule $\{k_s\}_{s=1}^{S}$, which progressively increases the number of filled tokens. For each selected position, the token is replaced by its most likely prediction, while the remaining positions remain masked for the next iteration.

This progressive strategy has two key benefits: (1) high-confidence tokens are fixed early, anchoring the sequence structure; (2) uncertain regions are deferred and refined in later iterations, reducing error accumulation and improving global coherence. After $S$ iterations, all tokens are committed, yielding the final generated sequence.

### 3.3 MOTION TOKENIZATION WITH RESIDUAL VQ

Following masked modeling approaches such as MoMask (Guo et al., 2024), we discretize continuous 3D motion sequences into discrete tokens via a *Residual Vector Quantizer (RVQ)*, providing higher representational capacity while maintaining stability. Given a motion $M \in \mathbb{R}^{T \times J \times 3}$, the encoder produces discrete indices:

$$z \in \{1, \ldots, N\}^{\lfloor \frac{T}{r} \rfloor \times R},$$

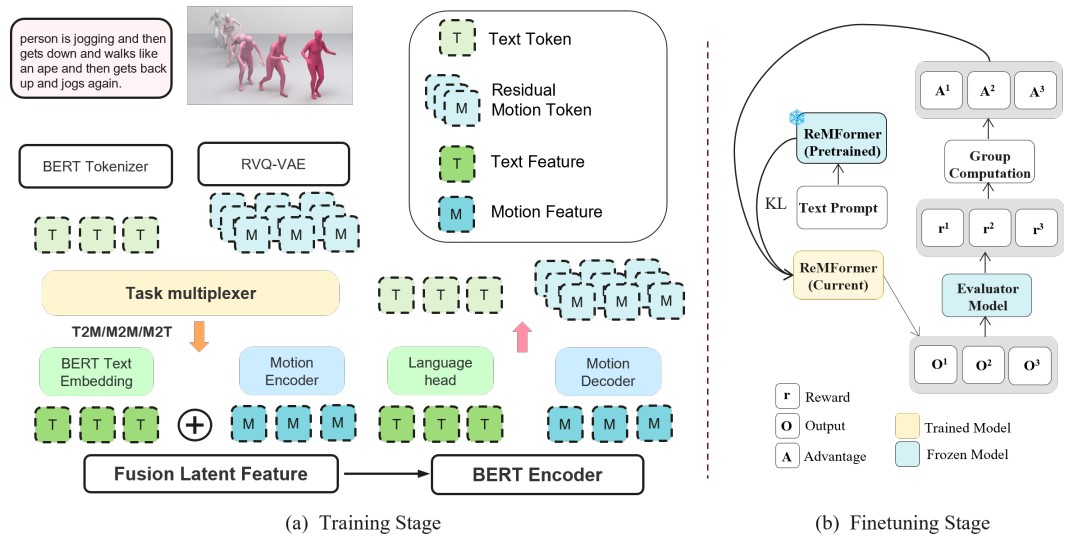

(a) Training Stage        (b) Finetuning Stage

Figure 2: Overview of MotionDDM. Our unified framework supports text-to-motion (T2M), motion-to-text (M2T), and motion-to-motion (M2M) tasks with RVQ-based motion tokenization, multi-task masked training, confidence-guided progressive inference, and GRPO fine-tuning.

---

**Algorithm 1** Confidence-Guided Progressive Inference

---

**Require:** Corrupted sequence $\tilde{y}$, condition $x$, inference steps $S$
**Ensure:** Generated sequence $y$
1: **for** $s = 1$ to $S$ **do**
2:     Predict logits $p_\theta(y_t \mid \tilde{y}, x)$ for all masked $t$
3:     Compute confidence $c_t = \max_v p_\theta(y_t = v \mid \tilde{y}, x)$
4:     Select top-$k_s$ masked positions with highest $c_t$
5:     Commit: $\tilde{y}_t \leftarrow \arg\max_v p_\theta(y_t = v \mid \tilde{y}, x)$
6: **end for**
7: **return** $y$

---

where $R$ is the number of residual layers. Each layer uses an independent codebook and $N$ is the number of entities of each codebook. $r$ is the ratio of temporal compression.

## 3.4 MODEL ARCHITECTURE

**MotionDDM** is designed as a unified denoising architecture for *text-to-motion (T2M)*, *motion-to-text (M2T)*, and *motion-to-motion (M2M)* generation. The model builds upon a bidirectional masked language model while introducing modality-specific encoders and decoders for motion. This hybrid design allows MotionDDM to preserve the strengths of pretrained LLMs for text reasoning while explicitly modeling the structure of motion tokens. An overview of the architecture is shown in Figure 2.

**Language backbone.** We adopt a pretrained **BERT-based masked language model** as the textual backbone. BERT's bidirectional contextualization naturally aligns with our discrete diffusion-style denoising objective, in contrast to autoregressive LLMs that impose a strict left-to-right order. The backbone produces contextual embeddings for textual tokens and serves as the fusion space where text and motion features interact. We extend the tokenizer with additional special tokens to represent motion masks and padding, ensuring that the model can process multimodal sequences consistently.

**Motion token encoder.** Continuous motion sequences are first discretized via Residual Vector Quantization (RVQ), yielding multi-level motion tokens. To embed these tokens into the LLM's hidden space, we introduce a dedicated motion encoder. For each RVQ codebook level, an embed-

ding table and a lightweight Transformer encoder process the token stream, capturing hierarchical motion dynamics. The outputs are fused through learnable weights and augmented with positional embeddings. This design enables motion features to be expressed in the same representational space as text tokens, while preserving high-fidelity motion details.

**Motion token decoder.**  Reconstruction of motion tokens is handled by a motion decoder. Each RVQ level is paired with a Transformer-based prediction head that refines backbone features and outputs logits over the motion vocabulary. By decoding tokens across RVQ levels in parallel, the decoder preserves coarse-to-fine motion fidelity while supporting efficient training and inference. An additional output projection layer maps hidden states to motion tokens, ensuring compatibility with the denoising objective.

## 3.5 REWARD DESIGN

To fine-tune MotionDDM beyond likelihood training, we design task-specific reward functions for *text-to-motion* (T2M) and *motion-to-text* (M2T), and optimize them jointly with GRPO (Sec. 3.6). The rewards are chosen to reflect semantic fidelity and modality-specific quality criteria, while remaining compatible with reinforcement learning.

**Reward for Text-to-Motion (T2M).**  Given a text prompt $x$, the model generates a motion $\hat{m}$. To evaluate its semantic correctness, we re-infer a pseudo caption $\hat{t}$ using MotionDDM's motion-to-text branch and compare it with the ground-truth caption $t$ in the CLIP embedding space:

$$R_{\text{T2M}}(\hat{m}, t) \;=\; \cos\!\big(\text{CLIP}(\hat{t}),\ \text{CLIP}(t)\big). \tag{3}$$

This reward captures whether generated motion conveys the intended textual meaning, without relying on heuristic metrics or external evaluators.

**Reward for Motion-to-Text (M2T).**  For a given motion $m$, the model outputs a caption $\hat{t}$. We combine verb consistency and semantic similarity, scaled by a length penalty (LP) to discourage degenerate outputs:

$$R_{\text{M2T}}(\hat{t}, t) = \Big[\lambda_{\text{verb}}\, \text{VerbMatch}(\hat{t}, t) + \lambda_{\text{clip}}\, \cos\!\big(\text{CLIP}(\hat{t}), \text{CLIP}(t)\big)\Big] \cdot \text{LP}(|\hat{t}|, |t|), \tag{4}$$

$$\text{LP}(|\hat{t}|, |t|) = \exp\!\Big(-\gamma\,\big|1 - \tfrac{|\hat{t}|}{\max(|t|, 1)}\big|\Big). \tag{5}$$

Here $\text{VerbMatch}$ measures verb overlap between $\hat{t}$ and $t$, CLIP similarity reflects semantic alignment, and LP penalizes captions that are disproportionately short or long. We use default weights $(\lambda_{\text{verb}}, \lambda_{\text{clip}}) = (0.3,\ 0.7)$ to emphasize more on semantic consistency and $\gamma = 1.0$.

## 3.6 GRPO OBJECTIVE

We adopt **Group Relative Policy Optimization (GRPO)** (Shao et al., 2024) to fine-tune MotionDDM. GRPO is a critic-free variant of PPO that eliminates the need for a value network by normalizing rewards within candidate groups, thereby reducing computation and memory costs.

**Objective.**  For each input $q$ (text or motion), we sample $G$ candidate outputs $\{o_i\}_{i=1}^{G}$ from the old policy $\pi_{\text{old}}$, compute rewards $r_i = R(o_i, q)$ (Sec. 3.5), and update the policy $\pi_\theta$ by maximizing:

$$J_{\text{GRPO}}(\theta) = \mathbb{E}_{q,\,o_i}\!\left[\frac{1}{G}\sum_{i=1}^{G}\min\!\Big(\rho_i(\theta)A_i,\ \text{clip}(\rho_i(\theta),\, 1-\epsilon,\, 1+\epsilon)\,A_i\Big)\right] - \beta\, D_{\text{KL}}\big(\pi_\theta(\cdot|q)\,\|\,\pi_{\text{ref}}(\cdot|q)\big), \tag{6}$$

Here $\rho_i(\theta) = \frac{\pi_\theta(o_i|q)}{\pi_{\text{old}}(o_i|q)}$ is the *importance ratio*, i.e., the likelihood of a sampled output under the current policy relative to the old policy. $A_i = \frac{r_i - \mu}{\sigma}$ is the group-normalized advantage, $\epsilon = 0.1$ is the clipping range, and $\beta = 0.004$ is the KL coefficient. We found these settings to yield stable training, while larger or smaller values often led to unstable updates.

Table 1: Quantitative results of Text-to-Motion and Motion-to-Text on HumanML3D.

| Category | Method | T2M | | | | | | | M2T | | | | | | |
|---|---|---|---|---|---|---|---|---|---|---|---|---|---|---|---|
| | | R@1↑ | R@2↑ | R@3↑ | FID↓ | Div→ | MM↑ | MM Dist↓ | R@1↑ | R@3↑ | BLEU@1↑ | BLEU@4↑ | ROUGE-L↑ | CIDEr↑ | BERTScore↑ |
| | Real Motion | 0.511 | 0.703 | 0.797 | 0.002 | 9.503 | - | 2.974 | 0.523 | 0.828 | - | - | - | - | - |
| T2M Only | MDM | - | - | 0.611 | 0.544 | 9.559 | 2.799 | 5.566 | - | - | - | - | - | - | - |
| | MotionDiffuse | 0.491 | 0.681 | 0.782 | 0.630 | 9.410 | 1.553 | 3.113 | - | - | - | - | - | - | - |
| | MLD | 0.481 | 0.673 | 0.772 | 0.473 | 9.724 | 2.413 | 3.196 | - | - | - | - | - | - | - |
| | MoMask | 0.521 | 0.713 | 0.807 | 0.045 | 9.620 | 1.241 | 2.958 | - | - | - | - | - | - | - |
| | T2M-GPT | 0.492 | 0.679 | 0.775 | 0.141 | 9.722 | 1.831 | 3.121 | - | - | - | - | - | - | - |
| | ReMoDiffuse | 0.510 | 0.698 | 0.795 | 0.103 | 9.018 | 1.795 | 2.974 | - | - | - | - | - | - | - |
| | MoGenTS | 0.529 | 0.719 | 0.812 | 0.033 | 9.570 | - | 2.867 | - | - | - | - | - | - | - |
| | MotionLCM | 0.502 | 0.698 | 0.798 | 0.304 | 9.607 | 2.259 | 3.012 | - | - | - | - | - | - | - |
| | ReMoMask | 0.531 | 0.722 | 0.813 | 0.099 | 9.535 | 2.823 | 2.865 | - | - | - | - | - | - | - |
| | MaskControl | - | - | 0.805 | 0.083 | 9.395 | - | - | - | - | - | - | - | - | - |
| Separated Model | TM2T | 0.424 | 0.618 | 0.729 | 1.501 | 8.589 | 2.424 | 3.467 | 0.516 | 0.823 | 48.9 | 7.0 | 38.1 | 16.8 | 32.2 |
| | LaMP | 0.557 | 0.751 | 0.843 | 0.032 | 9.571 | - | 2.759 | 0.547 | 0.831 | 47.8 | 13.0 | 37.1 | 28.9 | - |
| | MG-MotionLLM | 0.516 | 0.706 | 0.802 | 0.303 | 9.960 | 2.125 | 2.952 | 0.592 | 0.866 | - | 8.1 | - | - | 36.7 |
| Unified Model | MotionGPT | 0.492 | 0.681 | 0.778 | 0.232 | 9.528 | 2.008 | 3.096 | 0.543 | 0.827 | 48.2 | 12.5 | 37.4 | 29.2 | 32.4 |
| | MotionGPT2 | 0.427 | 0.627 | 0.764 | 0.614 | 11.256 | 2.357 | 3.164 | 0.558 | 0.838 | 48.7 | 13.8 | 37.6 | 29.8 | 32.6 |
| | MotionGPT3 | 0.543 | 0.735 | **0.828** | 0.217 | 9.662 | 1.366 | **2.793** | 0.573 | 0.858 | 51.1 | 8.4 | 38.7 | 10.4 | 30.3 |
| | MoTe | **0.548** | **0.737** | 0.825 | 0.075 | - | **2.399** | 2.867 | **0.577** | **0.871** | 46.7 | 11.2 | 37.4 | 31.5 | 30.3 |
| | Ours w/o GRPO | 0.528 | 0.723 | 0.818 | 0.050 | **9.515** | 2.016 | 2.867 | 0.569 | 0.850 | 63.9 | 22.6 | 47.0 | 57.2 | 37.5 |
| | Ours w/ GRPO | 0.528 | 0.724 | 0.818 | **0.047** | 9.419 | 2.000 | 2.862 | **0.577** | 0.855 | **64.2** | **22.7** | **47.1** | **58.1** | **37.7** |

# 4 EXPERIMENTS

## 4.1 IMPLEMENTATION DETAILS

Our model is trained on $16 \times 32$ GB Ascend 910 NPUs for 50 epochs with 32 mini-batch. Motion sequences are quantized into discrete tokens using a 6-layer RVQ tokenizer with 1024 codewords per layer. Text is tokenized with a pre-trained BERT-large tokenizer (Devlin et al., 2019). The backbone uses a hidden size of 1024 with 16 transformer layers. The downsample ratio $r$ is 4. Using a 20 fps frame rate, 1.5 tokens per frame, and a 1024-size codebook (10 bits/token), the resulting token rate is 30 tokens/s with a bitrate of 300 bits/s.

Optimization uses AdamW (Loshchilov & Hutter, 2019) with a learning rate of $5 \times 10^{-5}$, weight decay 0.01, and a linear warmup of 5k steps. Inference uses progressive unmasking with $K = 30$ refinement steps and classifier-free guidance (CFG) with a constant scale 3.0. GRPO fine-tuning is applied as a second stage (Sec. 3.6).

## 4.2 MAIN RESULTS

Table 1 reports quantitative comparisons on the HumanML3D (Guo et al., 2022a) benchmark. Overall, MotionDDM (w/o GRPO) achieves strong advantages in both generation quality and text-generation metrics: for T2M quality, MotionDDM attains a best-in-table FID of 0.050 (before we apply GRPO finetuning), and it also shows excellent semantic consistency and diversity-related scores. For M2T, MotionDDM leads by a large margin on text metrics (BLEU@1 = 63.9, BLEU@4 = 22.6, ROUGE-L = 47.0, CIDEr = 57.2, BERTScore = 37.6), indicating the model not only generates high-fidelity motions but also recovers their semantic descriptions accurately. With respect to retrieval, MotionDDM obtains R@1 = 0.569 and R@3 = 0.850, which are competitive with the strongest baselines (some baselines retain slight advantages on individual retrieval measures). In sum, MotionDDM delivers a very competitive trade-off across T2M (geometric/perceptual quality) and M2T (textual quality/semantic alignment) objectives. Without GRPO, MotionDDM already surpasses baselines; GRPO further improves both motion realism and caption quality.

## 4.3 ABLATION STUDIES

**Ablation — Backbone scaling and scaling law.** Table 3 compares three masked-language backbones (ALBERT, BERT-base, BERT-large) to study the correlation between model size and generation quality. We observe a monotonic improvement on T2M task with backbone scale (BERT-large > BERT-base > ALBERT), yielding consistent gains across key metrics such as R-Precision, FID. However, in terms of text performance, bert-base shows better results, likely because the tex-

Table 2: Quantitative results of Text-to-Motion and Motion-to-Text on KIT-ML.

| Method | T2M | | | | | | | M2T | | | | | | |
|---|---|---|---|---|---|---|---|---|---|---|---|---|---|---|
| | R@1↑ | R@2↑ | R@3↑ | FID↓ | Div→ | MM↑ | MM Dist↓ | R@1↑ | R@3↑ | BLEU@1↑ | BLEU@4↑ | ROUGE-L↑ | CIDEr↑ | BERTScore↑ |
| Real motion | 0.424 | 0.649 | 0.779 | 0.031 | 11.08 | - | 2.788 | 0.399 | 0.793 | - | - | - | - | - |
| TM2T | 0.280 | 0.463 | 0.587 | 3.599 | 9.473 | **3.292** | 4.591 | 0.359 | 0.668 | 46.7 | **18.4** | 44.2 | **79.5** | 23.0 |
| MotionGPT | 0.366 | 0.558 | 0.680 | 0.510 | 10.35 | 2.328 | 3.527 | - | - | - | - | - | - | - |
| MotionGPT2 | **0.427** | 0.627 | **0.764** | 0.614 | 11.256 | 2.357 | 3.164 | - | - | - | - | - | - | - |
| MoTe | 0.419 | **0.627** | 0.741 | 0.256 | - | 2.615 | 3.216 | **0.421** | **0.765** | 44.9 | 14.5 | 41.8 | 55.6 | 35.9 |
| Ours | 0.406 | 0.620 | 0.741 | **0.206** | **10.892** | 1.690 | **2.983** | 0.396 | 0.723 | **52.5** | 17.8 | **48.0** | 68.7 | **37.7** |

Table 3: Ablation study: backbone scaling and scaling law.

| Backbone | Text-to-Motion | | | | Motion-to-Text | | | | | | |
|---|---|---|---|---|---|---|---|---|---|---|---|
| | R@1↑ | R@2↑ | R@3↑ | FID↓ | R@1↑ | R@3↑ | BLEU@1↑ | BLEU@4↑ | ROUGE-L↑ | CIDEr↑ | BERTScore↑ |
| ALBERT | 0.455 | 0.647 | 0.753 | 0.947 | 0.528 | 0.821 | 63.1 | 22.5 | 46.1 | 54.5 | 34.4 |
| BERT-base | 0.493 | 0.684 | 0.780 | 0.135 | **0.582** | **0.857** | **65.0** | **23.4** | **47.6** | **58.9** | **37.5** |
| BERT-large | **0.528** | **0.723** | **0.818** | **0.050** | 0.569 | 0.850 | 63.9 | 22.6 | 47.0 | 57.2 | **37.5** |

tual corpus in HumanML3D is relatively limited. As future work, it is necessary to investigate the scaling laws on larger-scale motion generation datasets.

Table 4: Ablation on computation vs quality. Text metrics (BLEU@1, BLEU@4, ROUGE-L, CIDEr, BERTScore) are in percentage. Our model is evaluated under different denoising steps with BERT-large backbone.

| Method | Text-to-Motion | | | | | Motion-to-Text | | | | | | | Computational Cost | |
|---|---|---|---|---|---|---|---|---|---|---|---|---|---|---|
| | R@1↑ | R@2↑ | R@3↑ | FID↓ | Latency(s) | R@1↑ | R@3↑ | BLEU@1↑ | BLEU@4↑ | ROUGE↑ | BERTScore↑ | Latency(s) | #Params | FLOPs/sample |
| MotionGPT | 0.492 | 0.681 | 0.778 | 0.232 | 1.04 | 0.543 | 0.827 | 48.2 | 12.5 | 37.4 | 32.4 | 0.48 | 220M | 7.45T |
| MotionGPT-3 | **0.543** | **0.735** | **0.828** | 0.217 | 1.02 | 0.573 | 0.858 | 51.1 | 8.4 | 38.7 | 30.3 | 1.30 | 238M | 11T |
| MG-MotionLLM | 0.516 | 0.706 | 0.802 | 0.303 | 1.09 | **0.592** | **0.866** | - | 8.1 | - | 36.7 | 0.60 | 220M | 1.66T |
| Ours(5 steps) | 0.523 | 0.719 | 0.812 | 0.120 | 0.41 | 0.465 | 0.744 | 54.7 | 16.2 | 42.2 | 19.2 | 0.21 | 473M | 0.64T |
| Ours(10 steps) | 0.527 | 0.724 | 0.817 | 0.068 | 0.76 | 0.510 | 0.795 | 56.8 | 19.1 | 44.9 | 26.4 | 0.38 | 473M | 1.28T |
| Ours(20 steps) | 0.528 | 0.723 | 0.818 | **0.050** | 1.55 | 0.568 | 0.845 | 62.5 | 22.0 | **47.3** | 35.4 | 0.81 | 473M | 2.56T |
| Ours(30 steps) | 0.524 | 0.720 | 0.815 | 0.052 | 2.39 | 0.569 | 0.850 | **63.9** | **22.6** | 47.0 | **37.5** | 1.18 | 473M | 3.84T |

**Latency–quality trade-off.** The progressive denoising mechanism of discrete diffusion enables a smooth trade-off between inference latency and generation quality by tuning the number of sampling steps and confidence thresholds. We measure latency and core quality metrics (FID, BLEU, R@1, and human preference) under several sampling budgets (e.g., $K = 30, 20, 10$; see Table 4). Results indicate that reducing $K$ from 30 to 5 substantially cuts inference time while only incurring modest quality degradation. Despite having a larger model size than prior baselines, our model achieves a better quality–latency trade-off: very few steps (e.g., $K = 5$) already outperform existing methods in T2M FID with minimal latency, while more steps (10–30) further improve quality metrics at the cost of higher but still reasonable latency. This demonstrates that the number of diffusion steps provides flexible operating points balancing speed and quality beyond previous approaches.

**Effect of Masking Schedule.** We further ablate the effect of masking schedules for text and motion tokens. Table 5 compares the baseline configuration (linear schedule applied to both modalities) with two alternatives: (i) applying a cosine schedule to motion tokens while keeping text linear (mcos_linear), and (ii) applying cosine schedules to both text and motion tokens (mcos_mcos).

We observe that the baseline linear masking yields the best overall retrieval accuracy and lowest FID, while cosine masking significantly degrades motion fidelity, especially when applied jointly to text and motion. Interestingly, cosine-based schedules produce slightly higher diversity, suggesting that they encourage more varied predictions at the cost of consistency. These results indicate that linear masking provides a more stable learning signal for MotionDDM, and mixing schedules across modalities does not yield additional benefits.

**Effect of RVQ Stage Depth.** Table 6 shows that multi-stage residual quantization significantly improves reconstruction fidelity and downstream generation/understanding performance. A 6-stage RVQ consistently outperforms single-stage quantization on FID, R-Precision, and all text-based

Table 5: Ablation on masking schedules for text and motion tokens. Baseline uses linear masking for both modalities. We compare with cosine masking applied either to motion only (mcos_linear) or to both (mcos_mcos).

| Schedule | R@1↑ | R@2↑ | R@3↑ | FID↓ | Diversity→ |
|---|---|---|---|---|---|
| Baseline (linear + linear) | **0.528** | **0.723** | **0.818** | **0.050** | **9.300** |
| mcos_linear (motion=cosine, text=linear) | 0.502 | 0.699 | 0.795 | 0.369 | 9.030 |
| mcos_mcos (motion=cosine, text=cosine) | 0.507 | 0.701 | 0.798 | 0.264 | 8.947 |

Table 6: Ablation study: model performance under different RVQ layer.

| #RVQ Layer | Reconstruction | | Text-to-Motion | | | | Motion-to-Text | | | | | | |
|---|---|---|---|---|---|---|---|---|---|---|---|---|---|
| | MSE↓ | R@1↑ | R@2↑ | R@3↑ | FID↓ | R@1↑ | R@3↑ | BLEU@1↑ | BLEU@4↑ | ROUGE-L↑ | CIDEr↑ | BERTScore↑ |
| 1 | 0.0830 | 0.515 | 0.711 | 0.806 | 0.200 | 0.558 | 0.841 | 62.8 | 22.0 | 46.5 | 55.0 | 36.2 |
| 4 | 0.0215 | 0.527 | 0.722 | 0.821 | 0.076 | **0.579** | 0.845 | **64.2** | **22.8** | **47.2** | **58.3** | 37.0 |
| 6 | 0.0118 | 0.528 | 0.723 | 0.818 | **0.050** | 0.569 | **0.850** | 63.9 | 22.6 | 47.0 | 57.2 | **37.5** |
| 8 | **0.0101** | **0.531** | **0.729** | **0.824** | 0.121 | 0.573 | 0.847 | 63.8 | 22.3 | 47.0 | 57.4 | 37.1 |

metrics. The hierarchical residual codebooks allow the first stage to capture coarse, low-frequency body structure while later stages encode high-frequency details. This frequency-wise decomposition substantially reduces quantization error for fine details and provides the discrete diffusion backbone with more informative tokens, yielding motions that are clearer in both semantics and detail and producing more accurate textual descriptions. Reconstruction MSE decreases monotonically with increasing RVQ depth, confirming that deeper quantization better preserves motion details. Downstream T2M/M2T metrics improve up to 4–6 layers but saturate or fluctuate slightly beyond that, likely due to the trade-off between tighter reconstruction and more complex codebooks.

## 4.4 QUALITATIVE RESULTS

We provide qualitative examples to illustrate the characteristics of MotionDDM's bidirectional denoising paradigm. Unlike one-pass autoregressive generation, MotionDDM can iteratively refine outputs by re-masking and re-denoising subsequences. This flexibility enables progressive improvement of motion quality and the ability to adjust local segments without regenerating the entire sequence. The qualitative results can be found in supplementary materials.

Figure 3 shows a text-to-motion example: MotionDDM generates motions that follow the caption and preserve coherent body dynamics. Figure 4 shows a motion-to-text case: given a motion sequence, MotionDDM produces concise captions that capture key actions and their temporal order.

These results indicate that MotionDDM handles both directions within a unified framework. Compared with autoregressive approaches such as MotionGPT and MotionGPT3, MotionDDM tends to preserve fine-grained motion details throughout the sequence, maintaining later-stage semantics without relying on strong early-frame cues. Visual comparisons are provided in the submitted appendix material.

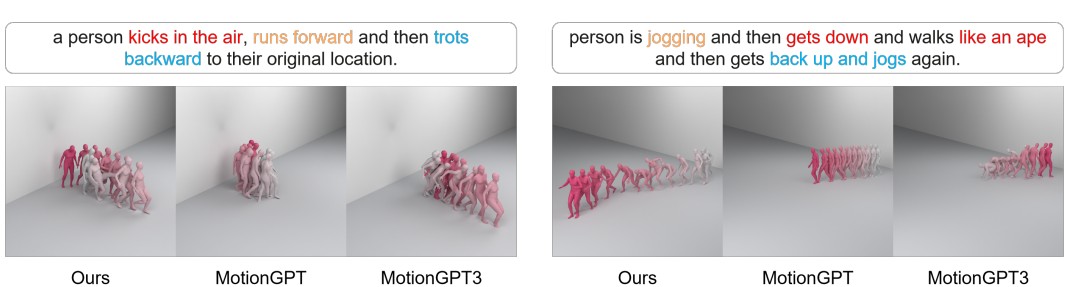

a person kicks in the air, runs forward and then trots backward to their original location.

person is jogging and then gets down and walks like an ape and then gets back up and jogs again.

Ours    MotionGPT    MotionGPT3    Ours    MotionGPT    MotionGPT3

Figure 3: Text-to-motion comparison: MotionDDM generates more coherent and semantically aligned motions.

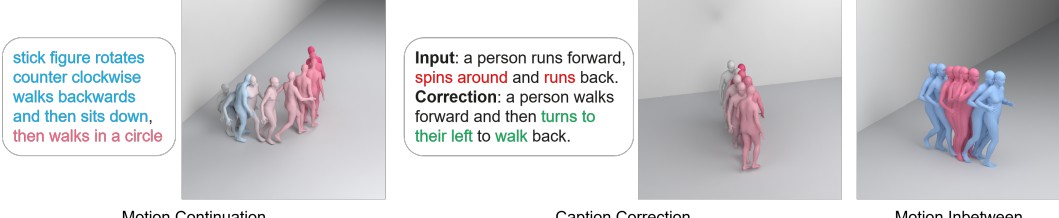

Figure 4: Motion-to-text comparison: MotionDDM produces concise and accurate action descriptions.

Motion Continuation          Caption Correction          Motion Inbetween

Figure 5: Application: Examples of downstream tasks enabled by MotionDDM

# 5 APPLICATIONS

## 5.1 MOTION COMPLETION

In motion inbetweening, the beginning and ending segments of a sequence are provided while the middle portion is masked. The model must reconstruct the missing part so that the whole sequence remains temporally coherent and kinematically natural. Unlike methods that rely on text, MotionDDM performs completion in a *text-free setting*, iteratively denoising the masked region to infer plausible dynamics. As shown in Figure 5 motion inbetween, MotionDDM refines the missing segment step by step, producing smooth transitions and semantically meaningful motions.

Beyond inbetweening, MotionDDM supports *motion continuation*, where a motion and its textual description are given and additional semantic content is appended to the text to extend the sequence. The model uses the new linguistic cues to generate successive motion segments, continuing the motion naturally. Figure 5 motion continuation demonstrates coherent extensions.

## 5.2 OTHER POTENTIAL APPLICATIONS

Beyond T2M, M2T, and motion completion, MotionDDM naturally extends to other tasks. It can perform motion-guided caption correction: when a caption mismatches the action (e.g., describing a walking motion as running), the model refines the text to match the observed behavior. An example is shown in Figure 5 caption correction, where iterative denoising progressively aligns predictions with temporal and semantic constraints.

These capabilities arise directly from MotionDDM's unified design, without modifying the architecture. More visualizations of these use cases can be found in the submitted appendix materials.

# 6 CONCLUSIONS

We introduced **MotionDDM**, a unified framework for text-to-motion and motion-to-text generation. The model combines masked progressive refinement with classifier-free guidance, enabling efficient bi-directional modeling. We further incorporated **GRPO fine-tuning**, which provides task-specific reinforcement signals and consistently improves semantic alignment and output quality across both directions. Experiments on HumanML3D show that MotionDDM achieves state-of-the-art results compared to previous unified motion-language models. In addition, the same framework extends naturally to motion completion and editing without architectural changes. Future work includes scaling beyond skeleton sequences to mesh-based or multi-agent motion, and exploring more robust reward functions for GRPO to further enhance controllability and quality.

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

# APPENDIX FOR MOTIONDDM

## REPRODUCIBILITY STATEMENT

We have made extensive efforts to ensure the reproducibility of MotionDDM. The datasets used in our experiments (HumanML3D) are publicly available. Details of our pipeline, motion encoder/decoder modules, and the BERT-based language backbone are presented in Sec. 3.4. Training procedures, including optimization settings, masking schedules, and reinforcement fine-tuning with GRPO, are documented in Sec. 3.6. Evaluation protocols and metrics for T2M, M2T, and motion completion will be discussed in details at Comprehensive descriptions of model architectures and hyperparameter settings are further provided in Appendix A.

## LLM USAGE

A large language model was used as a writing assistant to improve the clarity and readability of the manuscript. All final research decisions and methodological contributions were made and verified by the authors.

## A  EVALUATION METRICS

For **Text-to-Motion (T2M)**, follow existing works Guo et al. (2022b; 2024); Jiang et al. (2024); Wang et al. (2024); Zhu et al. (2025), we evaluate motion quality and text–motion alignment. Motion realism is measured by Fréchet Inception Distance (FID), while R-Precision (R@1/2/3) and Multimodal Distance (MM Dist) assess semantic consistency between motion and text. We further report Diversity (Div) to capture variation across generated motions, and Multi-Modality (MM) to measure the ability to produce multiple plausible outputs for the same text. For **Motion-to-Text (M2T)**, follow existing works Guo et al. (2022b; 2024); Jiang et al. (2024); Wang et al. (2024); Zhu et al. (2025), we adopt standard captioning metrics, including BLEU@1/4, ROUGE-L, CIDEr, and BERTScore, which jointly evaluate lexical overlap and semantic similarity to reference descriptions. In addition, we report R-Precision to measure alignment between generated texts and their corresponding motions. The detailed formulations of each metric are introduced below.

**R-Precision.** R-Precision measures retrieval performance by computing the fraction of relevant items within the top-$R$ retrieved results. In text-to-motion, this means retrieving the correct motion from a database given a text query, or vice versa.

$$R\text{-Prec} = \frac{|\text{Rel} \cap \text{Top-}R|}{R} \tag{7}$$

where Rel is the set of relevant items (ground-truth matches) and Top-$R$ is the set of retrieved items at rank $R$. The metric ranges from 0 to 1, with higher values indicating better retrieval accuracy.

**Fréchet Inception Distance (FID).** FID Guo et al. (2022a) measures the distributional distance between real and generated samples in a feature space, capturing both mean and covariance statistics. Lower FID indicates that generated samples are closer to real samples in distribution.

$$\text{FID} = \|\mu_r - \mu_g\|_2^2 + \text{Tr}\big(\Sigma_r + \Sigma_g - 2(\Sigma_r \Sigma_g)^{1/2}\big) \tag{8}$$

where $\mu_r, \Sigma_r$ are the mean and covariance of real samples in the feature space, and $\mu_g, \Sigma_g$ are the corresponding statistics of generated samples. The first term measures the distance between means, while the second term accounts for differences in covariance structure.

**Diversity.** Diversity Guo et al. (2022a) quantifies the average pairwise distance between generated samples in the feature space, reflecting variability among generated motions. Higher values indicate more diverse generations.

$$\text{Div} = \frac{2}{M(M-1)} \sum_{i<j} \|f(x_i) - f(x_j)\|_2 \tag{9}$$

where $\{x_i\}_{i=1}^M$ are generated samples and $f(\cdot)$ is a feature extractor such as a motion encoder. The summation averages the pairwise Euclidean distance across all distinct sample pairs.

**Multi-Modality.** Multi-Modality Guo et al. (2022a) measures the variability of outputs generated from the same input condition, capturing the model's ability to produce multiple plausible outputs. It is computed as the average pairwise distance between multiple samples generated for the same condition.

$$\text{MM} = \frac{2}{K(K-1)} \sum_{i<j} \|f(x_i^t) - f(x_j^t)\|_2 \tag{10}$$

where $\{x_i^t\}_{i=1}^K$ are the generated samples conditioned on the same text $t$, and $f(\cdot)$ is a feature extractor. The metric encourages both relevance to the condition and diversity among outputs.

**Multimodal Distance.** Multimodal Distance Guo et al. (2022a) evaluates how closely aligned text and motion representations are in a shared embedding space. Lower values indicate better alignment between modalities.

$$\text{MM Dist} = \frac{1}{N} \sum_{i=1}^N d(f_{\text{text}}(t_i), f_{\text{motion}}(m_i)) \tag{11}$$

where $\{(t_i, m_i)\}_{i=1}^N$ are paired text and motion samples, $f_{\text{text}}$ and $f_{\text{motion}}$ are embedding functions for text and motion respectively, and $d(\cdot, \cdot)$ is a distance function such as Euclidean distance or cosine distance.

**BLEU.** BLEU Papineni et al. (2002) evaluates the n-gram precision of generated text against reference text, penalized by a brevity term to avoid favoring short outputs. The most common setup is BLEU@4, which considers up to 4-gram matches.

$$\text{BLEU@N} = \text{BP} \cdot \exp\left(\frac{1}{N} \sum_{n=1}^N \log p_n\right) \tag{12}$$

where $N$ is the maximum n-gram order (commonly $N = 4$), and $p_n$ is the modified n-gram precision defined as

$$p_n = \frac{\sum_{\text{ngram} \in C} \min\left(\text{Count}_C(\text{ngram}), \max_{R \in \text{Refs}} \text{Count}_R(\text{ngram})\right)}{\sum_{\text{ngram} \in C} \text{Count}_C(\text{ngram})}. \tag{13}$$

The brevity penalty (BP) is applied to discourage very short candidates:

$$\text{BP} = \begin{cases} 1 & \text{if } c > r, \\ e^{(1-r/c)} & \text{if } c \leq r, \end{cases} \tag{14}$$

where $C$ denotes the candidate sentence, Refs the set of reference sentences, $\text{Count}_C(\text{ngram})$ the number of times an n-gram appears in the candidate, and $\text{Count}_R(\text{ngram})$ the count of the same n-gram in reference $R$. The clipping operation $\min(\cdot)$ ensures that the n-gram precision does not reward repeated n-grams beyond what occurs in references. Here $c$ is the length (in tokens) of the candidate sentence, and $r$ is the effective reference length, chosen as the reference length closest to $c$.

**ROUGE-L.** ROUGE-L Lin (2004) measures the quality of generated text by computing the longest common subsequence (LCS) between candidate and reference, which captures sentence-level fluency without requiring consecutive matches. The score combines precision and recall of the LCS using an $F$-measure formulation.

$$\text{ROUGE-L} = \frac{(1 + \beta^2) \cdot P_{LCS} \cdot R_{LCS}}{R_{LCS} + \beta^2 \cdot P_{LCS}} \tag{15}$$

where $R_{LCS} = \frac{LCS(c,r)}{|r|}$ and $P_{LCS} = \frac{LCS(c,r)}{|c|}$, with $LCS(c,r)$ denoting the length of the longest common subsequence between candidate $c$ and reference $r$. The parameter $\beta$ controls the relative importance of recall and precision (commonly $\beta = 1$ for equal weighting). Here $|c|$ is the candidate length in tokens and $|r|$ is the reference length in tokens.

**CIDEr.** CIDEr Vedantam et al. (2015) evaluates the similarity of a candidate sentence against multiple references using a TF-IDF weighted n-gram representation. It is designed to capture consensus among reference captions and reduce the effect of common n-grams.

$$\text{CIDEr}(c, S) = \frac{1}{|S|} \sum_{s \in S} \frac{g(c) \cdot g(s)}{\|g(c)\|\|g(s)\|} \tag{16}$$

where $c$ is the candidate sentence, $S$ is the set of reference sentences, and $g(x)$ denotes the TF-IDF vector representation of n-grams extracted from text $x$. The numerator $g(c) \cdot g(s)$ is the dot product between the candidate and reference vectors, while the denominator normalizes by their Euclidean norms to compute cosine similarity.

**BERTScore.** BERTScore Zhang et al. (2019) evaluates the semantic similarity between generated and reference sentences using contextual embeddings from a pretrained language model such as BERT. It computes the average maximum similarity between tokens across candidate and reference.

$$\text{BERTScore}(c, r) = \frac{1}{|c|} \sum_{x \in c} \max_{y \in r} \cos\big(f(x), f(y)\big) \tag{17}$$

where $c$ is the candidate sentence, $r$ is the reference sentence, $f(\cdot)$ is the contextual embedding function from BERT, and $\cos(\cdot, \cdot)$ denotes cosine similarity. For each token $x$ in the candidate, the most similar token $y$ in the reference is found in embedding space, and the similarities are averaged across all candidate tokens.

# B    MORE QUANTITATIVE RESULTS

## B.1    EFFECT OF MULTI-TASK PROPORTION

Table 7 examines how different training mixtures of text-to-motion (T2M), motion-to-text (M2T), and motion-only samples influence performance. We observe that M2T results are largely invariant to the data ratio: BLEU, ROUGE, CIDEr, and BERTScore remain stable even when the proportion of captioning samples is reduced. This indicates that the pretrained language backbone already provides strong priors for text generation, such that extensive exposure to captioning data is not required for competitive M2T performance.

In contrast, T2M quality exhibits greater sensitivity to the task mixture. Allocating excessive capacity to M2T leads to a measurable decline in motion fidelity (lower recall, higher FID), reflecting the higher difficulty of modeling discrete motion tokens compared to text. The 8:1:1 configuration yields the most favorable balance, achieving strong language grounding while preserving the highest motion generation quality.

## B.2    EFFECT OF CLASSIFIER-FREE GUIDANCE (CFG) AT INFERENCE

We further study the impact of classifier-free guidance (CFG) scale on inference quality. Table 8 reports results across scales $\{2, 3, 4, 5, 6\}$. We find that retrieval accuracy (R@1/2/3) improves when

Table 7: Ablation study: model performance under different data mixture ratios.

| Text-to-Motion (%) | Motion (%) | Motion-to-Text (%) | Text-to-Motion | | | | Motion-to-Text | | | | | | |
|---|---|---|---|---|---|---|---|---|---|---|---|---|---|
| | | | R@1↑ | R@2↑ | R@3↑ | FID↓ | R@1↑ | R@3↑ | BLEU@1↑ | BLEU@4↑ | ROUGE-L↑ | CIDEr↑ | BERTScore↑ |
| 8 | 1 | 1 | **0.528** | **0.723** | 0.818 | **0.050** | 0.569 | **0.850** | 63.9 | **22.6** | 47.0 | 57.2 | 37.6 |
| 7 | 1 | 2 | 0.523 | 0.722 | **0.820** | 0.091 | **0.578** | **0.850** | 64.4 | 22.3 | **47.1** | **58.2** | **38.1** |
| 6 | 1 | 3 | 0.515 | 0.712 | 0.810 | 0.077 | 0.553 | 0.825 | 60.4 | 19.6 | 44.4 | 53.0 | 35.8 |

increasing the scale from 2 to 4, with $s = 4$ yielding the highest recall. FID is lowest around $s = 3$–4, while larger scales ($s \geq 5$) noticeably degrade fidelity. Diversity also reaches its maximum at $s = 3$, whereas the matching score remains relatively stable across different settings.

Overall, moderate CFG values ($s = 3$–4) provide the best trade-off, enhancing semantic alignment without sacrificing motion quality, while overly strong guidance harms realism and reduces diversity.

Table 8: Ablation on classifier-free guidance (CFG) scale during inference. Moderate scales ($s = 3$–4) achieve the best balance between motion fidelity and semantic alignment.

| CFG Scale | R@1↑ | R@2↑ | R@3↑ | FID ↓ |
|---|---|---|---|---|
| 2 | 0.520 | 0.714 | 0.811 | 0.063 |
| 3 (Ours) | **0.528** | 0.723 | **0.818** | **0.050** |
| 4 | **0.528** | **0.725** | 0.817 | 0.059 |
| 5 | 0.525 | 0.721 | **0.818** | 0.067 |
| 6 | 0.526 | 0.720 | 0.814 | 0.112 |

## B.3 Effect of Down-weighting [PAD] Tokens

We found that the BERT backbone tends to over-generate [PAD] tokens at inference time. To mitigate this, we down-weight their probability by a multiplicative factor. Table 9 compares different settings. A too aggressive penalty (0.7) harms caption quality, leading to lower BLEU, ROUGE, CIDEr, and BERTScore. In contrast, a mild penalty (0.8) achieves the best results, substantially improving caption fluency and semantic alignment while maintaining retrieval performance. This demonstrates that carefully tuning the [PAD] suppression factor is crucial for balancing sequence validity and output quality.

*Note:* Results reported in the appendix may vary slightly from the main tables due to differences in experimental runs and random seeds.

## B.4 Effect of Multi-task Training (M2M / T2M / M2T)

To study the effect of the multi-task formulation, we perform an ablation over the three training objectives: (i) Motion-to-Motion (M2M), (ii) Text-to-Motion (T2M), and (iii) Motion-to-Text (M2T). We toggle each objective on/off and report the full set of T2M and M2T metrics on HumanML3D. Results are summarized in Table 10.

The results show that the three objectives are mutually beneficial. Training only with T2M improves the T2M branch but yields weaker motion semantics, while training only with M2T produces stronger captioning but degrades motion quality. Introducing M2M further enhances motion coherence and enables effective classifier-free guidance. The complete configuration (M2M + T2M + M2T) achieves the best balance on both T2M and M2T metrics, indicating that unified optimization encourages better motion–text alignment and motion structure.

Table 9: Ablation on down-weighting the probability of generating `[PAD]` tokens during text inference. We compare multiplicative factors of 0.7 and 0.8 against the baseline (0.85).

| Pad prob factor | R@1 | R@3 | BLEU@1 | BLEU@4 | ROUGE-L | CIDEr | BERTScore |
|---|---|---|---|---|---|---|---|
| 0.70 | 0.571 | 0.849 | 44.9 | 12.2 | 38.3 | 29.9 | 29.7 |
| 0.80 | **0.585** | **0.861** | **61.7** | **20.6** | **46.0** | **56.3** | **37.1** |
| 0.85 (baseline) | 0.558 | 0.842 | 60.4 | 19.5 | 44.4 | 52.9 | 35.8 |

Table 10: **Multi-task ablation on HumanML3D.** Removing any single objective degrades either T2M or M2T performance, while the full configuration (M2M + T2M + M2T) achieves the best overall trade-off across metrics.

| Tasks | | | T2M Retrieval ↑ | | | T2M Generation | | | | M2T Retrieval ↑ | | M2T Text ↑ | | | | |
|---|---|---|---|---|---|---|---|---|---|---|---|---|---|---|---|---|
| M2M | T2M | M2T | R@1 | R@2 | R@3 | FID ↓ | Div → | MM ↑ | MM Dist ↓ | R@1 | R@3 | BLEU@1 | BLEU@4 | ROUGE-L | CIDEr | BERTScore |
| No | No | Yes | – | – | – | – | – | – | – | 0.547 | 0.822 | 56.9 | 16.3 | 42.8 | 45.9 | 34.3 |
| No | Yes | No | 0.518 | 0.704 | 0.801 | 0.323 | 9.335 | 2.014 | 2.999 | – | – | – | – | – | – | – |
| No | Yes | Yes | **0.530** | **0.725** | **0.824** | 0.114 | 9.583 | **2.399** | **2.839** | 0.561 | 0.849 | **65.3** | **23.2** | **47.2** | **57.6** | 37.4 |
| Yes | No | Yes | – | – | – | – | – | – | – | 0.559 | 0.838 | 56.4 | 15.9 | 42.5 | 47.1 | 34.5 |
| Yes | Yes | No | 0.515 | 0.714 | 0.814 | 0.174 | 9.146 | 2.106 | 2.956 | – | – | – | – | – | – | – |
| Yes | Yes | Yes | 0.528 | 0.723 | 0.818 | **0.050** | **9.515** | 2.016 | 2.867 | **0.569** | **0.850** | 63.9 | 22.6 | 47.0 | 57.2 | **37.5** |

