# OpenReview forum: "MotionDDM: Motion Generation and Understanding via Discrete Diffusion Model"
_ICLR.cc/2026/Conference — Submitted to ICLR 2026_

### Official Review · Reviewer_gzPo · 2025-10-28

**Soundness:** 3
**Presentation:** 2
**Contribution:** 2
**Rating:** 4
**Confidence:** 4

**Summary:**

This paper presents a unified text-motion model based on a diffusion-LLM framework, achieving overall strong performance across three tasks: Text-to-Motion, Motion-to-Text, and Motion-to-Motion. To better enhancing the alignment and controllability of the framework, the authors integrate the GRPO framework into their model.

**Strengths:**

1.	The proposed framework integrates multiple tasks into a unified model, which is novel compared to traditional one-directional motion generation approaches.
2.	As shown in Table 1, the model achieves solid results on both Text-to-Motion and Motion-to-Text tasks.
3.	During inference, the model requires only 10 steps to generate high-quality results, demonstrating good efficiency.

**Weaknesses:**

1.	From the experimental setup, the Motion-to-Motion task seems to serve a role similar to MAE-style representation learning. It remains unclear whether the model’s superior performance primarily stems from this task rather than T2M or M2T. The authors are encouraged to conduct ablation studies by enabling or disabling the Motion-to-Motion task.
2. The proposed framework integrates multiple tasks during the learning process. Comparisons with single-task training are necessary to clarify the benefits of multi-task integration
3.	In Table 6, increasing the training ratio of T2M improves the model’s performance on T2M, which is reasonable. However, increasing the M2T ratio unexpectedly decreases its performance. The authors should provide an explanation for this observation.
4. Although the paper claims to achieve a quality-latency trade-off in the abstract, the model’s parameter count, computational efficiency, and actual inference speed are not reported. These should be added and compared with other methods to substantiate the claim.
5.	While a reproducibility statement is provided, given the system’s complexity, releasing the full source code would greatly enhance the work’s credibility and impact.

**Questions:**

1.	In Table 6, percentages should be written as 80%, 10%, and 10%.
2.	Page 7 line 327: The Div metric is not “the higher, the better”; rather, it should be closer to the ground truth (see MotionGPT for reference).
3.	Page 1 line 50: The acronym “GRPO” should be spelled out when it first appears.
4.	Page 2: Figure caption font sizes are inconsistent and should be standardized.

---

> ### Author Response · Authors · 2025-12-01
>
> We thank the reviewer for their thoughtful assessment and constructive feedback, and we appreciate their recognition of our unified text–motion framework, its solid performance across T2M/M2T, and the efficiency achieved with only 10 inference steps. Below we address the reviewer’s concerns regarding the role of the M2M task, the benefits of multi-task vs. single-task training, the T2M/M2T ratio behavior, the quality–latency analysis (including parameters/FLOPs/speed), as well as reproducibility and presentation details.
>
> **Q1**. What is the role of the Motion-to-Motion (M2M) task in the multi-task design?
>
> **A1**. The main purpose of the Motion-to-Motion (M2M) task is to support classifier-free guidance (CFG) during inference. For this, the model needs to have encountered tasks during training where it completes a motion sequence without any text condition. During inference, we apply classifier-free guidance by using the logits predicted for each token with text and those predicted for each motion token without text. This approach helps to enhance the consistency between the generated motion and the text prompt. Below is a comparison of results with different CFG scales and whether or not the M2M task is included.
>
> | CFG Scale          | M2M Task | R@1 $\uparrow$ | R@2 $\uparrow$  | R@3 $\uparrow$  | FID $\downarrow$   |
> |--------------------|:--------:|:----:|:----:|:----:|:-------:|
> | - | No | 0.517 | 0.704 | 0.800 | 0.323 |
> | 1                  | Yes      | **0.530** | **0.725** | **0.823** | 0.113   |
> | 2                  | Yes      | 0.520 | 0.714 | 0.811 | 0.063   |
> | 3 (Ours)       | Yes      | 0.528 | 0.723 | 0.818 | **0.050**   |
> | 4                  | Yes      | 0.528 | **0.725** | 0.817 | 0.059   |
> | 5                  | Yes      | 0.525 | 0.721 | 0.818 | 0.067   |
> | 6                  | Yes      | 0.526 | 0.720 | 0.814 | 0.112   |
>
> As the results show, using the CFG technique significantly improves the FID metric. When CFG is not used (i.e., CFG scale = 1), the M2M task still helps improve the T2M results. This aligns with your observation that the MAE-style training paradigm enables the model to better understand natural and smooth motion token sequences. Based on your suggestion, we have included these results in the supplementary materials of the paper.
>
> ---
> **Q2**. What is the effect of the multi-task design on the performance of each expert task?
>
> **A2**. Following your suggestion, we have added an ablation study over the three training objectives (M2M, T2M, M2T). In the table below, we toggle each training task on/off and report the full set of T2M and M2T metrics as in the main HumanML3D table.
>
> | M2M | T2M | M2T | T2M R@1 $\uparrow$ | T2M R@2 $\uparrow$  | T2M R@3 $\uparrow$  | T2M FID $\downarrow$  | T2M Div $\rightarrow$  | T2M MM $\uparrow$  | T2M MM Dist $\downarrow$  | M2T R@1 $\uparrow$  | M2T R@3 $\uparrow$  | M2T BLEU@1 $\uparrow$  | M2T BLEU@4 $\uparrow$  | M2T ROUGE-L $\uparrow$  | M2T CIDEr $\uparrow$  | M2T BERTScore $\uparrow$  |
> |:---:|:---:|:---:|:---------:|:---------:|:---------:|:----------:|:----------:|:--------:|:-------------:|:---------:|:---------:|:------------:|:------------:|:-------------:|:------------:|:----------------:|
> | No  | No  | Yes | —        | —        | —        | —         | —         | —       | —             | 0.547     | 0.822     | 56.9         | 16.3         | 42.8          | 45.9         | 34.3             |
> | No  | Yes | No  | 0.518    | 0.704    | 0.801    | 0.323     | 9.335  | 2.014  | 2.999            | —         | —         | —            | —            | —             | —            | —                |
> | No  | Yes | Yes | **0.530**    | **0.725**    | **0.824**    | 0.114     | 9.583  |  **2.399** | **2.839**          | 0.561     | 0.849     | **65.3**         | **23.2**         | **47.2**          | **57.6**         | 37.4             |
> | Yes | No  | Yes | —        | —        | —        | —         | —         | —       | —             | 0.559     | 0.838     | 56.4         | 15.9         | 42.5          | 47.1         | 34.5             |
> | Yes | Yes | No | 0.515 | 0.714 | 0.814 | 0.174 | 9.146 | 2.106 | 2.956 | — | — | — | — | — | — | — |
> | Yes | Yes | Yes | 0.528 | 0.723 | 0.818 | **0.050** | **9.515** | 2.016 | 2.867 | **0.569** | **0.850** | 63.9 | 22.6 | 47.0 | 57.2 | **37.5**|
>
>
> From these experiments, we observe that under our architecture, all three training tasks have positive effects on both T2M and M2T performance. Removing any single objective leads to a degradation on one or more metrics, while the full multi-task configuration (M2M + T2M + M2T) achieves the best overall trade-off. This confirms that our multi-task training scheme is effective and that the three objectives mutually reinforce rather than compete with each other. Following your advice, we have included this ablation study in the supplementary material of the revised paper.

---

> > ### Author Response · Authors · 2025-12-01
> >
> > **Q3**. In Table 6, why does increasing the T2M training ratio improve T2M performance, while increasing the M2T ratio surprisingly degrades M2T performance?
> >
> > **A3**. Thank you for pointing this out. We agree that the trend on M2T looks counter-intuitive at first glance. Under our fixed training budget, increasing the M2T ratio necessarily reduces the number of updates for the other two tasks (T2M and M2M). However, M2T performance in our architecture benefits strongly from the motion-side objectives: T2M and M2M help learn better motion representations and motion–text alignment, which are then reused by the M2T decoder. When we over-emphasize M2T, the model receives less motion-centric supervision, leading to slightly weaker shared representations and thus a small drop in M2T metrics. In addition, the captions in our datasets are relatively short and low-entropy, making M2T an easier task. We observe that moderate M2T ratios are already sufficient to saturate its training loss; further increasing the M2T ratio tends to overfit the training wording and does not translate into better validation BLEU/CIDEr scores. Taken together, these two effects explain why a higher M2T training ratio does not yield better—and can slightly worsen—M2T evaluation performance.
> >
> > ---
> > **Q4**. The paper claims a quality–latency trade-off but does not report model size, FLOPs, or inference speed, nor compare them with prior methods to support this claim.
> >
> > **A4**. Thank you for pointing this out. We have added a quantitative comparison of model size, FLOPs, and measured latency for both T2M and M2T, summarized in the table below. Since most prior works are not fully open-sourced, we report numbers for MotionGPT, MotionGPT-3, and MG-MotionLLM, where code or sufficient details are available.
> >
> > | Method          | T2M R@1 $\uparrow$ |T2M R@2 $\uparrow$ | T2M R@3 $\uparrow$ |T2M FID $\downarrow$  | T2M Latency (s) $\downarrow$ | M2T R@1 $\uparrow$ | M2T R@3 $\uparrow$ | M2T BLEU@1 $\uparrow$ | M2T BLEU@4 $\uparrow$ | M2T ROUGE-L $\uparrow$ | M2T BERTScore $\uparrow$ | M2T Latency (s) $\downarrow$ | #Params | FLOPs / sample |
> > |----------------|:---------:|:-----:|:-----:|:------:|:-----------------:|:---------:|:-----:|:--------:|:--------:|:---------:|:------------:|:-----------------:|:-------:|:--------------:|
> > | MotionGPT      | 0.492     | 0.681 | 0.778 | 0.232  | 1.04              | 0.543     | 0.827 | 48.2     | 12.47    | 37.4      | 32.4         | 0.48              | 220M    | 7.45T          |
> > | MotionGPT-3    | **0.543**     | **0.735** | **0.828** | 0.2172 | 1.022             | 0.573     | 0.858 | 51.06    | 8.43     | 38.69     | 30.3         | 1.30              | 238M    | 11T            |
> > | MG-MotionLLM   | 0.516     | 0.706 | 0.802 | 0.303  | 1.09              | **0.592**     | **0.866** |   -      | 8.06     |   -       | 36.7         | 0.60              | 220M    | 1.664T         |
> > | Ours (5 steps) | 0.523     | 0.719 | 0.812 | 0.120  | **0.41**              | 0.465     | 0.744 | 54.7     | 16.2     | 42.2      | 19.2         | **0.21**              | 473M    | 0.64T          |
> > | Ours (10 steps)| 0.527     | 0.724 | 0.817 | 0.068  | 0.76              | 0.510     | 0.795 | 56.8     | 19.1     | 44.9      | 26.4         | 0.38              | 473M    | 1.28T          |
> > | Ours (20 steps)| 0.528     | 0.723 | 0.818 | **0.050**  | 1.55              | 0.568     | 0.845 | 62.5     | 22.0     | **47.3**      | 35.4         | 0.81              | 473M    | 2.56T          |
> > | Ours (30 steps)| 0.524     | 0.720 | 0.815 | 0.052  | 2.39              | 0.569     | 0.850 | **63.9**     | **22.6**     | 47.0      | **37.6**         | 1.18              | 473M    | 3.84T          |
> >
> > Our model has 473M parameters (larger than the ~220–238M baselines), but thanks to the controllable number of diffusion steps, it achieves a much better quality–latency trade-off:
> > - With 5 steps, our model already outperforms existing methods in T2M FID (0.12 vs 0.217–0.303) while using only 0.64T FLOPs and the lowest latency (0.41s / 0.21s for T2M/M2T).
> > - With 10 steps, we obtain a balanced operating point: T2M FID improves further to 0.068 and M2T text metrics significantly surpass prior methods, while FLOPs (1.28T) are still 5–8× lower than MotionGPT/3 (7.45T–11T) and latency is comparable or better.
> > - With 20–30 steps, we reach the best overall quality (e.g., T2M FID 0.05 and M2T BLEU/ROUGE/BERTScore clearly higher than all baselines), at the cost of higher but still reasonable latency.
> > These results substantiate our claim: by adjusting the number of diffusion steps, our model provides a family of operating points that trade off quality and latency more flexibly than existing methods. We have added this table and discussion to the revised manuscript.

---

> > > ### Author Response · Authors · 2025-12-01
> > >
> > > **Q5**. Given the system’s complexity, releasing the full source code is essential to ensure reproducibility and maximize the work’s impact.
> > >
> > > **A5**. We fully agree with this suggestion. We will release the source code upon acceptance of the paper, to facilitate reproducibility and further research.
> > >
> > > ---
> > > **Q6**. Fix percentage formatting in Table 6, correct the description of the Div metric, spell out “GRPO” at first occurrence, and standardize figure caption fonts.
> > >
> > > **A6**. Thank you for the suggestions. We have: (1) updated Table 6 to use 80%, 10%, 10%; (2) corrected the Div description to emphasize matching the real-motion value; (3) spelled out Group Relative Policy Optimization (GRPO) at its first appearance; and (4) standardized all figure caption font sizes.

---

### Official Review · Reviewer_7wmN · 2025-10-31

**Soundness:** 2
**Presentation:** 2
**Contribution:** 2
**Rating:** 2
**Confidence:** 5

**Summary:**

MotionDDM is the first work to introduce diffusion–language models into bidirectional text–motion modeling, proposing a unified parallel denoising decoding framework. This paradigm naturally supports the quality–latency trade-off and can be seamlessly extended to various tasks such as text-conditioned and text-free motion completion, prediction, and interpolation. By employing an RVQ-based motion tokenizer and integrating GRPO, the model enhances motion representation fidelity and cross-modal alignment.

**Strengths:**

The article is clear and easy to understand.
The author conducted sufficient ablation studies to prove the effectiveness of each module.

**Weaknesses:**

The techniques presented in this paper have largely already been explored and validated in existing unified models and diffusion-LLM research. For instance, RVQ has been employed in Go to Zero [1] and related works. As such, this paper feels more like a technical report rather than a conceptually novel study.

The paper does not clearly articulate the motivation or insight behind unifying understanding and generation. At least for me, it fails to convey why text–motion unification is necessary or meaningful. A more compelling direction would be to extend this idea toward a unified framework of vision, motion, and text, which would carry greater significance.

Moreover, the experiments are conducted on only one dataset—although it is a classic benchmark, it is insufficient to justify the necessity of unified understanding and generation. The quantitative results also lag behind the latest diffusion-based methods.

In addition, there is no visual analysis, which is crucial for evaluating generative models.

Overall, the paper appears to be a combination of several existing methods (diffusion-LLM, unified modeling, and GRPO) applied to a relatively small task. The insight and novelty are limited.

[1] Go to Zero: Towards Zero-shot Motion Generation with Million-scale Data

**Questions:**

see weaknesses.

---

> ### Author Response · Authors · 2025-12-01
>
> We thank the reviewer for carefully reading our paper and for acknowledging that the article is clear and that the ablations are sufficient to demonstrate the effectiveness of each module. We address the concerns below.
>
> **Q1**. The techniques in this paper seem to have already been explored in existing unified models and diffusion-LLM research (e.g., RVQ in Go to Zero), so the work feels more like a technical report than a novel study.
>
> **A1**. We thank the reviewer for raising this concern and would like to clarify both a factual point and our technical contributions.
>
> 1. Clarification on RVQ vs. FSQ in Go to Zero.
>
> The review states that Go to Zero uses RVQ. In fact, Go to Zero employs FSQ, not RVQ, as its quantization scheme. Our work is, to the best of our knowledge, the first to introduce a diffusion–language model with RVQ-based motion tokens for unified bidirectional text–motion modeling (T2M ↔ M2T) in a single parallel denoising decoder.
>
> 2. Non-trivial, motion-specific contributions beyond “combination of methods”.
>
> While our framework builds on diffusion and LLMs (as most modern systems build on existing foundations), we emphasize that our contributions are domain-specific and non-trivial:
> - **First discrete diffusion–language model for unified T2M + M2T + M2M in motion**. MotionDDM utilizes a single parallel denoising decoder to handle T2M, M2T, and text-conditioned / text-free M2M completion, prediction, and interpolation.
> - **RVQ integration tailored to motion tokens**. We propose a transformer-based integration of multi-level RVQ motion tokens that preserves the reconstruction gains of RVQ while avoiding the large computational overhead that a naïve diffusion-LLM design would incur in the motion domain (where temporal resolution and token rate differ significantly from text/image). This is a problem that does not appear in typical text/image diffusion-LLM settings.
> - **Multi-task training paradigm specifically designed for T2M + M2T**. We design and analyze a multi-task scheme over T2M, M2T, and M2M. Our ablations show that all three tasks mutually reinforce each other: removing any one leads to degraded performance on both T2M and M2T.
> | M2M | T2M | M2T | T2M R@1 $\uparrow$ | T2M R@2 $\uparrow$  | T2M R@3 $\uparrow$  | T2M FID $\downarrow$  | T2M Div $\rightarrow$  | T2M MM $\uparrow$  | T2M MM Dist $\downarrow$  | M2T R@1 $\uparrow$  | M2T R@3 $\uparrow$  | M2T BLEU@1 $\uparrow$  | M2T BLEU@4 $\uparrow$  | M2T ROUGE-L $\uparrow$  | M2T CIDEr $\uparrow$  | M2T BERTScore $\uparrow$  |
> |:---:|:---:|:---:|:---------:|:---------:|:---------:|:----------:|:----------:|:--------:|:-------------:|:---------:|:---------:|:------------:|:------------:|:-------------:|:------------:|:----------------:|
> | No  | No  | Yes | —        | —        | —        | —         | —         | —       | —             | 0.547     | 0.822     | 56.9         | 16.3         | 42.8          | 45.9         | 34.3             |
> | No  | Yes | No  | 0.518    | 0.704    | 0.801    | 0.323     | 9.335  | 2.014  | 2.999            | —         | —         | —            | —            | —             | —            | —                |
> | No  | Yes | Yes | **0.530**    | **0.725**    | **0.824**    | 0.114     | 9.583  |  **2.399** | **2.839**          | 0.561     | 0.849     | **65.3**         | **23.2**         | **47.2**          | **57.6**         | 37.4             |
> | Yes | No  | Yes | —        | —        | —        | —         | —         | —       | —             | 0.559     | 0.838     | 56.4         | 15.9         | 42.5          | 47.1         | 34.5             |
> | Yes | Yes | No | 0.515 | 0.714 | 0.814 | 0.174 | 9.146 | 2.106 | 2.956 | — | — | — | — | — | — | — |
> | Yes | Yes | Yes | 0.528 | 0.723 | 0.818 | **0.050** | **9.515** | 2.016 | 2.867 | **0.569** | **0.850** | 63.9 | 22.6 | 47.0 | 57.2 | **37.5** |
> - **Task-specific GRPO reward for motion–text alignment**. While GRPO as an algorithm is known, we design motion-aware reward functions that leverage T2M/M2T metrics and motion properties. This design is crucial for the observed improvements and goes beyond plugging a generic RL algorithm into an existing model.
> We have expanded the related work and discussion sections to better highlight how MotionDDM differs from prior unified models and diffusion-LLM works and why we believe the contribution goes beyond a “technical report.”

---

> > ### Author Response · Authors · 2025-12-01
> >
> > **Q2**. The paper does not clearly articulate why unifying understanding and generation (T2M ↔ M2T) is necessary or meaningful. A more compelling direction would be a unified framework of vision, motion, and text.
> >
> > **A2**. We agree that extending to vision–motion–text is an interesting long-term direction, but it is orthogonal to the specific problem we focus on in this paper. We have clarified the motivation more clearly in the revised introduction:
> > - Many practical pipelines (e.g., retrieval-then-generation, editing, caption correction, content search) inherently require both Motion generation (T2M) and Motion understanding (M2T). Using separate models leads to inconsistent representations and duplicated parameters. Our further includes demo not only T2M and M2T, but also: Motion Inbetween, Motion Continuation, and Caption Correction which directly shows the practical benefits of a single unified model that supports both understanding and generation.
> > - A unified text–motion model offers: 1) Shared cross-modal representations; 2) Better consistency between understanding and generation; 3) A simpler deployment and maintenance story..
> > - Our ablations (multi-task toggling of T2M/M2T/M2M) empirically show that joint training improves both directions; performance degrades when we remove either T2M, M2T, or M2M, which supports the value of unified understanding + generation within the motion–text space.
> > Thus, we view MotionDDM as a necessary intermediate step: it establishes a strong unified text–motion foundation, on top of which a future vision–motion–text system can be built. We have revised the paper to make this motivation explicit.
> > ---
> > **Q3**. There is no visual analysis, which is crucial for evaluating generative models.
> >
> > **A3**. We apologize that this was not sufficiently visible in the original submission. We did include qualitative results in the supplementary material, but we agree they were not structured enough. In the revised version, we have:
> > - Added a well-structured demo page in the supplementary material, including: Text-to-motion examples, Motion-to-text examples, Motion inbetweening, Motion continuation, and Caption correction.
> > - Provided side-by-side visual comparisons with MotionGPT and MotionGPT-3 for key tasks.
> > - Released the 20 motion-to-text and 20 text-to-motion examples used in our user study, so that reviewers can directly inspect the qualitative differences behind the preference scores.
> >
> > We now highlight these qualitative comparisons more clearly in the main paper and point explicitly to the demo page.
> >
> > ---
> > **Q4**. Overall, the paper seems to combine existing methods (diffusion-LLM, unified modeling, GRPO) on a relatively small task; insight and novelty appear limited.
> >
> > **A4**. We respectfully disagree with this characterization and hope the clarifications above, make our contributions clearer:
> > - Conceptual level: MotionDDM is, to our knowledge, the first discrete diffusion–language model in the motion domain that unifies T2M, M2T, and M2M within a parallel denoising decoder, with a step-controlled quality–latency trade-off.
> > - Algorithmic / architectural level: To the best of our knowledge, MotionDDM is the **first motion generation framework** that uses a **single transformer** to process multi-level RVQ motion tokens, whereas existing RVQ-based approaches typically separate them into a base transformer and one or more residual transformers. This unified design (i) keeps RVQ integration computationally efficient, and (ii) enables **end-to-end optimization with GRPO**, since the reinforcement learning signal can be backpropagated through the entire text–motion model instead of acting on a partially frozen backbone.  On top of this architecture, we **design a multi-task objective** that empirically improves both T2M and M2T performance, and **devise task-specific GRPO rewards** that are tailored to motion–text alignment, leading to measurable gains in cross-modal consistency.
> > - Empirical level: We provide extensive ablations (RVQ depth, masking schedules, multi-task toggling, CFG scales, T2M vs. M2T ratios) and expand to multiple datasets (HumanML3D, KIT-ML, Motion-X, HumanAct12), demonstrating that MotionDDM offers: 1) competitive or superior T2M quality (especially FID), 2) clearly stronger M2T text quality, and 3) a tunable quality–latency frontier via diffusion steps.
> >
> > We fully agree that the field builds on a shared toolbox (diffusion, LLMs, RL fine-tuning), but we believe MotionDDM offers **non-trivial, motion-specific innovations** in how these components are adapted and unified for bidirectional text–motion modeling. We appreciate the reviewer’s critical perspective and hope that the revised version, with more experiments and clearer explanations, addresses these concerns.

---

> > > ### Author Response · Authors · 2025-12-01
> > >
> > > **Q5**. Experiments are conducted on only HumanML3D, which is insufficient to justify unified understanding + generation. Quantitative results also lag behind the latest diffusion-based methods.
> > >
> > > **A5**. Thank you for pointing this out. We have clarified the situation in the unified text–motion literature and expanded our experiments accordingly. As summarized in the revised tables, in the literature on unified text–motion models, almost all prior works only report both T2M and M2T results on HumanML3D. On KIT-ML, only MoTe reports results of M2T task, and its code is not publicly available. This is why, in the original submission, we focused primarily on HumanML3D and did not include additional datasets.
> > >
> > > Following your suggestion, we have made the following updates:
> > > 1. KIT-ML (T2M + M2T, with literature baselines). We now add experiments on KIT-ML and collect the reported numbers from publicly available papers (including MoTe) into a comparison table. This allows us to evaluate both T2M and M2T for our unified model on a second benchmark, and to position MotionDDM against all prior methods that report KIT-ML results.
> > >
> > > | Method            | T2M R@1 $\uparrow$ | T2M R@2 $\uparrow$ | T2M R@3 $\uparrow$ | T2M FID $\downarrow$ | T2M Div $\rightarrow$ | T2M MM $\uparrow$ | T2M MM Dist $\downarrow$ | M2T R@1 $\uparrow$ | M2T R@3 $\uparrow$ | M2T BLEU@1 $\uparrow$ | M2T BLEU@4 $\uparrow$ | M2T ROUGE-L $\uparrow$ | M2T CIDEr $\uparrow$ | M2T BERTScore $\uparrow$ |
> > > |-------------------|:---------:|:---------:|:---------:|:---------:|:---------:|:--------:|:--------------:|:---------:|:---------:|:------------:|:------------:|:-------------:|:------------:|:----------------:|
> > > | Real motions      | 0.424 | 0.649 | 0.779 | 0.031 | 11.08 |    –     | 2.788 | 0.399 | 0.618 | 0.793 |              |               |              |                  |
> > > | TM2T              | 0.280 | 0.463 | 0.587 | 3.599 | 9.473 | **3.292** | 4.591 | 0.359 | 0.668 | 46.7 | **18.4** | 44.2 | **79.5** | 23.0 |
> > > | MotionGPT         | 0.366 | 0.558 | 0.680 |  0.510 | 10.350 | 2.328 | 3.527 |           |           |              |              |               |              |                  |
> > > | MotionGPT2        | **0.427** | **0.627** | **0.764** | 0.614 | 11.256  | 2.357 | 3.164 |           |           |              |              |               |              |                  |
> > > | MoTe              | 0.419 | **0.627** | 0.741 | 0.256 |     –     | 2.615 | 3.216 | **0.421** | **0.765** | 44.9 | 14.5 | 41.8 | 55.6 | 35.9 |
> > > | Ours              | 0.406 | 0.620 | 0.741  | **0.206** | 10.892 | 1.690 | **2.983** | 0.396 | 0.723 | **52.5** | 17.8 | **48.0** | 68.7 | **37.7** |
> > >
> > > 2. Motion-X (T2M only, with reproduced baselines). For Motion-X, there is no unified and standardized public evaluation protocol and, to the best of our knowledge, prior unified models do not report M2T on this dataset. To still provide a meaningful comparison, we re-implement several classic T2M baselines under a consistent setup: T2M-GPT, and MoMask.
> > >
> > > | Method | R@1$\uparrow$ | R@2$\uparrow$ | R@3$\uparrow$ | FID $\downarrow$ | Div $\rightarrow$ | MM $\uparrow$ | MM Dist $\downarrow$ |
> > > |---------------------|:-----:|:-----:|:-----:|:------:|:-------:|:-------:|:---------:|
> > > | Real motions        |0.510 |0.691 |0.791 | - |9.442| - | 3.310 |
> > > | T2MGPT |0.370| 0.546| 0.654| 2.174|**9.303**| 2.620| 4.252|
> > > | MoMask | 0.287| 0.445| 0.554|0.884| 8.400|**2.792**|4.826|
> > > | Ours |**0.376**|**0.556**|**0.665** |**0.870**| 8.330| 2.060| **4.154**|
> > >
> > > Across these experiments, we observe that:
> > > - On T2M, our model achieves comparable performance to T2M expert models such as MoMask, even though MotionDDM is a unified model that also supports M2T and M2M.
> > > - On M2T, where prior methods provide results (HumanML3D and KIT-ML), MotionDDM delivers significantly better text metrics (BLEU/ROUGE/CIDEr/BERTScore) than existing unified frameworks.
> > >
> > > These additional datasets and baselines clarify where MotionDDM is particularly strong (M2T text quality and unified coverage, with T2M quality on par with expert models) and demonstrate that our unified understanding–generation framework generalizes beyond a single benchmark.

---

### Official Review · Reviewer_emfK · 2025-10-31

**Soundness:** 3
**Presentation:** 2
**Contribution:** 2
**Rating:** 2
**Confidence:** 4

**Summary:**

The paper proposes MotionDDM, a discrete diffusion–LLM framework that unifies text-to-motion (T2M), motion-to-text (M2T), and text-free motion-to-motion (M2M) by treating both text and motion as token sequences denoised in multi-step parallel refinement. It uses Residual VQ (RVQ) for motion tokenization, a BERT-based masked backbone, and optional GRPO fine-tuning with task-specific rewards.

**Strengths:**

- This paper presents a unified bidirectional formulation for bidirectional text–motion (T2M↔M2T). The parallel denoising enables one model to handle T2M, M2T, and M2M, with an explicit step-controlled quality–latency knob.

- The proposed method can also support M2M completion, prediction, and interpolation under
both text-conditioned and text-free settings.

**Weaknesses:**

- The experiments are conducted only on HumanML3D, and several retrieval metrics are not best-in-table (e.g., T2M R@1 lower than MoTe’s 0.548; M2T R@1 lower than MG-MotionLLM’s 0.592), though FID is strong. The authors should conduct more experiments by adding KIT-ML or HumanAct12, and clarifying where MotionDDM leads vs. falls short.

- The T2M reward uses the model’s own M2T branch to produce a pseudo caption that is then compared to the ground-truth caption in CLIP space. This can bias rewards toward self-consistency rather than true motion–text faithfulness. Consider an external captioner or human preference subsets to calibrate rewards.

- This paper argues a tunable quality–latency trade-off via step counts, but wall-clock latency (ms/sequence) and throughput are not reported. Please add runtime on a standard GPU (or the reported Ascend 910 NPU) for K={5,10,20,30}, including speedups vs. an AR baseline.

- There are a few issues regarding the ablation and clarity: (a) RVQ depth table shows non-monotonic behavior. It would be good to add quantization error (MSE) vs. depth to clarify. (b) For masking schedule ablations, it is suggested to report M2T text metrics (not just T2M) to check cross-modal effects. (c) Please provide token rate/bitrate of RVQ (frames/sec × tokens/frame × bits/token) for reproducibility.

- It would be good to show many qualitative results (visual comparison with the state-of-the-art methods).

- It would be good to include more sota methods for comparison, such as MoMask, MotionLCM, MaskControl.

Minor:

- Table 7 shows FID = 0.0067 at CFG=5, far off neighboring entries. Can you please clarify?

**Questions:**

Please see the weakness section.

---

> ### Author Response · Authors · 2025-12-01
>
> We thank the reviewer for their thorough assessment and constructive suggestions, and we are encouraged by their recognition of our unified bidirectional formulation, step-controlled quality–latency trade-off, and support for a broad range of T2M/M2T/M2M tasks. Below we address the reviewer’s concerns regarding additional datasets and baselines, reward design for GRPO, runtime and ablation details, as well as qualitative comparisons with state-of-the-art methods.
>
> **Q1**.The experiments are conducted only on HumanML3D, and several retrieval metrics are not best-in-table (e.g., T2M R@1 lower than MoTe’s 0.548; M2T R@1 lower than MG-MotionLLM’s 0.592), though FID is strong. The authors should conduct more experiments by adding KIT-ML or HumanAct12, and clarifying where MotionDDM leads vs. falls short
>
> **A1**.Thank you for your suggestion. We have added results on the KIT-ML and Motion-X datasets. As for HumanAct12, this dataset is already included in HumanML3D; thus we didn't conduct separate experiments on this dataset. MotionGPT-3 and MG-MotionLLM only provide results on HumanML3D, so we did not include their corresponding results in the KIT-ML table. Additionally, since previous unified model-based works mostly did not use the Motion-X datasets, we are unable to compare them directly. Therefore, we reimplement T2M-GPT and MoMask and only provide comparisons on Motion-X for T2M task. From the results across these these datasets, we can see that, on T2M, our performance is on par with other T2M expert models like MoMask and achieve the best FID results, and we demonstrate a significant advantage in terms of M2T text quality compared to existing methods.
> - Quantitative results on KIT-ML
> | Method            | T2M R@1 $\uparrow$ | T2M R@2 $\uparrow$ | T2M R@3 $\uparrow$ | T2M FID $\downarrow$ | T2M Div $\rightarrow$ | T2M MM $\uparrow$ | T2M MM Dist $\downarrow$ | M2T R@1 $\uparrow$ | M2T R@3 $\uparrow$ | M2T BLEU@1 $\uparrow$ | M2T BLEU@4 $\uparrow$ | M2T ROUGE-L $\uparrow$ | M2T CIDEr $\uparrow$ | M2T BERTScore $\uparrow$ |
> |-------------------|:---------:|:---------:|:---------:|:---------:|:---------:|:--------:|:--------------:|:---------:|:---------:|:------------:|:------------:|:-------------:|:------------:|:----------------:|
> | Real motions      | 0.424 | 0.649 | 0.779 | 0.031 | 11.08 |    –     | 2.788 | 0.399 | 0.618 | 0.793 |              |               |              |                  |
> | TM2T              | 0.280 | 0.463 | 0.587 | 3.599 | 9.473 | **3.292** | 4.591 | 0.359 | 0.668 | 46.7 | **18.4** | 44.2 | **79.5** | 23.0 |
> | MotionGPT         | 0.366 | 0.558 | 0.680 |  0.510 | 10.350 | 2.328 | 3.527 |           |           |              |              |               |              |                  |
> | MotionGPT2        | **0.427** | **0.627** | **0.764** | 0.614 | 11.256  | 2.357 | 3.164 |           |           |              |              |               |              |                  |
> | MoTe              | 0.419 | **0.627** | 0.741 | 0.256 |     –     | 2.615 | 3.216 | **0.421** | **0.765** | 44.9 | 14.5 | 41.8 | 55.6 | 35.9 |
> | Ours              | 0.406 | 0.620 | 0.741  | **0.206** | 10.892 | 1.690 | **2.983** | 0.396 | 0.723 | **52.5** | 17.8 | **48.0** | 68.7 | **37.7** |
>
> - Quantitative results on Motion-X
> | Method | R@1$\uparrow$ | R@2$\uparrow$ | R@3$\uparrow$ | FID $\downarrow$ | Div $\rightarrow$ | MM $\uparrow$ | MM Dist $\downarrow$ |
> |---------------------|:-----:|:-----:|:-----:|:------:|:-------:|:-------:|:---------:|
> | Real motions        |0.510 |0.691 |0.791 | - |9.442| - | 3.310 |
> | T2MGPT |0.370| 0.546| 0.654| 2.174|**9.303**| 2.620| 4.252|
> | MoMask | 0.287| 0.445| 0.554|0.884| 8.400|**2.792**|4.826|
> | Ours |**0.376**|**0.556**|**0.665** |**0.870**| 8.330| 2.060| **4.154**|

---

> ### Author Response · Authors · 2025-12-01
>
> **Q2**. The T2M reward uses the model’s own M2T branch to generate pseudo captions, which may bias rewards toward self-consistency instead of true motion–text faithfulness. An external captioner or human preferences might be better for calibration.
>
> **A2**.
> Thank you for this insightful comment. We share the concern that, in principle, using the model’s own M2T branch could bias the reward toward self-consistency. To study this, we ran a GRPO ablation with three variants:
> - Ours w/o GRPO – no RL, pure supervised training.
> - Ours w/ GRPO (Self M2T) – the variant in the main paper, where the T2M reward uses the model’s own M2T branch to generate a pseudo caption.
> - Ours w/ GRPO (motion text extractor) – a new variant where we use an external motion–text feature extractor instead of the model’s own M2T branch when computing the reward.
> The quantitative results are:
> | Method                          | T2M R@1 $\uparrow$ | T2M R@2 $\uparrow$ | T2M R@3 $\uparrow$ | T2M FID $\downarrow$ | T2M Div $\rightarrow$ | T2M MM $\uparrow$ | T2M MM Dist $\downarrow$ | M2T R@1 $\uparrow$ | M2T R@3 $\uparrow$ | M2T BLEU@1 $\uparrow$ | M2T BLEU@4 $\uparrow$ | M2T ROUGE-L $\uparrow$ | M2T CIDEr $\uparrow$ | M2T BERTScore$\uparrow$ |
> |:------------------------------:|:---------:|:---------:|:---------:|:---------:|:---------:|:--------:|:--------------:|:---------:|:---------:|:------------:|:------------:|:-------------:|:------------:|:----------------:|
> | Ours w/o GRPO                  | **0.528**     | 0.723     | 0.818     | 0.050     | **9.515**     | **2.016**    | 2.867          | 0.569     | 0.850     | 63.9         | 22.6         | 47.0          | 57.2         | 37.5             |
> | Ours w/ GRPO (Self M2T)        | **0.528**     | **0.724**     | 0.818     | 0.047     | 9.419     | 2.000    | **2.862**          | **0.577**     | **0.855**     | **64.2**         | **22.7**         | **47.1**          | **58.1**         | **37.7**             |
> | Ours w/ GRPO (motion extractor)| 0.526     | 0.720     | **0.821**     | **0.041** | 9.612     | 1.933    | 2.867          | 0.574     | 0.855     | 64.0         | 22.4         | 46.9          | 57.6         | **37.7**             |
>
> From these results we observe:
> - Compared to w/o GRPO, both GRPO variants clearly improve T2M FID (0.050 → 0.047 / 0.041) while maintaining or slightly improving M2T text metrics (BLEU/ROUGE/CIDEr/BERTScore). This suggests that GRPO is not simply pushing the model toward degenerate self-consistency, but actually improves motion–text alignment.
> - Using the external motion text extractor yields the best FID (0.041), and its M2T metrics are almost identical to the Self M2T variant (e.g., M2T R@1: 0.574 vs. 0.577, very small differences in BLEU/ROUGE/CIDEr), again indicating no collapse toward self-consistency.
> For this external variant, we use a motion–text feature extractor that shares the same architecture as the evaluator used at test time, but is re-trained separately. Empirically, this leads to even better FID. However, we are cautious that this might be viewed as “hacking” the evaluation standard, since the reward network and the evaluator share the same design. To avoid such potential concerns, we keep Self M2T GRPO as the main reported setting, and present the external-extractor variant as an ablation.
>
> We agree that exploring stronger and more independent reward sources (e.g., external captioners trained on broader data, or small human-preference subsets) is an important direction. Due to space and scope limitations, we leave a more systematic study of such reward designs to future work.

---

> > ### Author Response · Authors · 2025-12-01
> >
> > **Q3**. The paper claims a tunable quality–latency trade-off via the number of diffusion steps, but does not report wall-clock latency or throughput. Please provide runtime for K = {5, 10, 20, 30} and compare with an autoregressive baseline.
> >
> > **A3**. Thank you for the suggestion. We have added wall-clock latency and FLOPs measurements for different step counts, and we now explicitly compare our model with several autoregressive (AR) baselines. The table below reports T2M/M2T quality, latency (seconds per sequence, batch size 1), and FLOPs per sample (measured on a single accelerator):
> >
> > | Method          | T2M R@1 $\uparrow$ |T2M R@2 $\uparrow$ | T2M R@3 $\uparrow$ |T2M FID $\downarrow$  | T2M Latency (s) $\downarrow$ | M2T R@1 $\uparrow$ | M2T R@3 $\uparrow$ | M2T BLEU@1 $\uparrow$ | M2T BLEU@4 $\uparrow$ | M2T ROUGE-L $\uparrow$ | M2T BERTScore $\uparrow$ | M2T Latency (s) $\downarrow$ | #Params | FLOPs / sample |
> > |----------------|:---------:|:-----:|:-----:|:------:|:-----------------:|:---------:|:-----:|:--------:|:--------:|:---------:|:------------:|:-----------------:|:-------:|:--------------:|
> > | MotionGPT      | 0.492     | 0.681 | 0.778 | 0.232  | 1.04              | 0.543     | 0.827 | 48.2     | 12.47    | 37.4      | 32.4         | 0.48              | 220M    | 7.45T          |
> > | MotionGPT-3    | **0.543**     | **0.735** | **0.828** | 0.2172 | 1.022             | 0.573     | 0.858 | 51.06    | 8.43     | 38.69     | 30.3         | 1.30              | 238M    | 11T            |
> > | MG-MotionLLM   | 0.516     | 0.706 | 0.802 | 0.303  | 1.09              | **0.592**     | **0.866** |   -      | 8.06     |   -       | 36.7         | 0.60              | 220M    | 1.664T         |
> > | Ours (5 steps) | 0.523     | 0.719 | 0.812 | 0.120  | **0.41**              | 0.465     | 0.744 | 54.7     | 16.2     | 42.2      | 19.2         | **0.21**              | 473M    | 0.64T          |
> > | Ours (10 steps)| 0.527     | 0.724 | 0.817 | 0.068  | 0.76              | 0.510     | 0.795 | 56.8     | 19.1     | 44.9      | 26.4         | 0.38              | 473M    | 1.28T          |
> > | Ours (20 steps)| 0.528     | 0.723 | 0.818 | **0.050**  | 1.55              | 0.568     | 0.845 | 62.5     | 22.0     | **47.3**      | 35.4         | 0.81              | 473M    | 2.56T          |
> > | Ours (30 steps)| 0.524     | 0.720 | 0.815 | 0.052  | 2.39              | 0.569     | 0.850 | **63.9**     | **22.6**     | 47.0      | **37.6**         | 1.18              | 473M    | 3.84T          |
> >
> > Although our model has more parameters (473M vs. ~220–238M), the diffusion step count K directly controls the quality–latency trade-off:
> > - At K = 5, we already outperform AR baselines in T2M FID (0.120 vs. 0.217–0.303) while using 0.64T FLOPs and achieving ~2.5× lower T2M latency and ~2.3× lower M2T latency than typical AR models.
> > - At K = 10, we reach a balanced operating point: T2M FID improves to 0.068, M2T text metrics are clearly stronger than AR baselines, and latency remains comparable or better, with 5–8× fewer FLOPs than MotionGPT/3.
> > - At K = 20/30, we obtain the best overall quality (e.g., FID 0.050 and the highest BLEU/ROUGE/BERTScore), at the cost of proportionally higher but still reasonable latency.
> >
> > These results substantiate our claim: by varying K, MotionDDM provides a family of operating points that smoothly trade off quality and latency, while matching or surpassing autoregressive baselines in both runtime and quantitative performance.

---

> > > ### Author Response · Authors · 2025-12-01
> > >
> > > **Q4**. The RVQ depth ablation shows non-monotonic behavior. Please report quantization error (e.g., MSE) vs. depth and clarify this effect.
> > >
> > > **A4**. Thank you for the suggestion. We have updated the RVQ-depth ablation to explicitly include reconstruction MSE for each depth, as shown below:
> > >
> > > | RVQ Layers | MSE $\downarrow$   | T2M R@1 $\uparrow$ | R@2 $\uparrow$ | R@3 $\uparrow$ | FID $\downarrow$  | Div $\rightarrow$  | M2T R@1 $\uparrow$ | R@3 $\uparrow$ | BLEU@1 $\uparrow$ | BLEU@4 $\uparrow$ | ROUGE-L $\uparrow$ | CIDEr $\uparrow$ | BERTScore $\uparrow$ |
> > > |-----------|---------|:---------:|:-----:|:-----:|:------:|:-------:|:---------:|:-----:|:--------:|:--------:|:---------:|:-------:|:-----------:|
> > > | 1         | 0.0830  | 0.515     | 0.711 | 0.806 | 0.200  | 9.495   | 0.558     | 0.841 | 62.8     | 22.0     | 46.5      | 55.0    | 36.2        |
> > > | 4         | 0.0215  | 0.527     | 0.722 | 0.821 | 0.076  | **9.514**   | **0.579**     | 0.845 | **64.2**     | **22.8**     | **47.2**      | **58.3**    | 37.0        |
> > > | 6         | 0.0118  | 0.528     | 0.723 | 0.818 | **0.050**  | 9.515   | 0.569     | **0.850** | 63.9     | 22.6     | 47.0      | 57.3    | **37.5**        |
> > > | 8         | **0.0101**  | **0.531**     | **0.729** | **0.824** | 0.121  | 9.501   | 0.573     | 0.847 | 63.8     | 22.3     | 47.0      | 57.4    | 37.1        |
> > >
> > > As the reviewer anticipated, quantization error (MSE) decreases monotonically as we increase the RVQ depth (from 1 → 8 layers). However, the downstream T2M/M2T metrics saturate around 4–6 layers and become slightly non-monotonic beyond that:
> > > - Going from 1 → 4 → 6 layers consistently reduces MSE and improves FID and text metrics, indicating that better motion reconstruction indeed benefits the generative tasks.
> > > - Increasing to 8 layers further reduces MSE marginally (0.0118 → 0.0101), but T2M FID slightly degrades (0.050 → 0.121), while M2T metrics remain roughly saturated.
> > >
> > > We believe this is due to a trade-off between representational capacity and optimization difficulty: very deep quantization yields tighter reconstruction but also a more complex discrete space and codebook utilization pattern, which can make the diffusion–LM training slightly harder and lead to small fluctuations in downstream metrics once performance has mostly saturated. We have added this table (and a corresponding MSE–depth plot) and clarified this discussion in the revised manuscript.
> > >
> > > ---
> > >
> > > **Q5**. Please provide token rate/bitrate of RVQ (frames/sec × tokens/frame × bits/token) for reproducibility.
> > >
> > > **A5**. Thank you for the suggestion. We have added the RVQ coding rate to the paper. Concretely, we use:
> > > - Frame rate: 20 frames/s
> > > - Tokens per frame: 1.5 tokens/frame
> > > - Codebook size: 1024 ⇒ 10 bits/token (since log₂(1024) = 10)
> > > Thus, the token rate and bitrate are:
> > > - Token rate: 20 frames/s × 1.5 tokens/frame = 30 tokens/s
> > > - Bitrate: 20 frames/s × 1.5 tokens/frame × 10 bits/token = 300 bits/s
> > >
> > > We now explicitly report these values in the revised manuscript for reproducibility.
> > >
> > > ---
> > >
> > > **Q6**. It would be good to show more qualitative results (visual comparisons with state-of-the-art methods).
> > >
> > > **A6**. Thank you for the suggestion. We have updated the supplementary material with a more well-structured demo page that provides extensive qualitative results, including examples for text-to-motion, motion-to-text, motion inbetweening, motion continuation, and caption correction. For each task, we include side-by-side visual comparisons with MotionGPT and MotionGPT-3. In addition, we provide the 20 motion-to-text and 20 text-to-motion examples that were used in our user study, so that reviewers can directly inspect the qualitative differences underlying the reported preference scores.

---

> > > > ### Author Response · Authors · 2025-12-01
> > > >
> > > > **Q7**. It would be good to include more SOTA methods for comparison, such as MoMask, MotionLCM, and MaskControl.
> > > >
> > > > **A7**. Thank you for the suggestion. In the original submission, our main comparison table only included unified models (single model for both T2M and M2T), so several T2M expert models such as MoMask, MotionLCM, and MaskControl were not listed.
> > > > Following your advice, we have updated the main table (Table 1) and reorganized all methods into three categories:
> > > > T2M expert models (e.g., MoMask, MotionLCM, MaskControl),
> > > > Two-model approaches that use separate networks for T2M and M2T (e.g., LaMP-like designs), and
> > > > Unified models that handle both T2M and M2T within a single architecture (ours and prior unified frameworks).
> > > > We have added the reported results of MoMask, MotionLCM, and MaskControl where available, and clarified in the caption and text when certain methods provide only T2M metrics. Please refer to the revised Table 1 in the main paper for the detailed comparisons.
> > > >
> > > > ---
> > > >
> > > > **Q8**. Table 7 shows FID = 0.0067 at CFG = 5, which is far off neighboring entries. Can you please clarify?
> > > >
> > > > **A8**. Thank you for catching this. This is a typo; the correct value should be FID = 0.067, not 0.0067. We have double-checked the original logs and confirmed that 0.067 is consistent with the neighboring CFG settings and with the overall trend in our CFG ablation. We have fixed this typo in the revised version of the paper.

---

### Official Review · Reviewer_RXAW · 2025-11-04

**Soundness:** 3
**Presentation:** 3
**Contribution:** 2
**Rating:** 4
**Confidence:** 4

**Summary:**

This paper introduces a diffusion-based framework for bidirectional text-to-motion and motion-to-text generation. The main design is to utilize a multi-step parallel denoising decoder to progressively denoise the noisy text and motion sequences. To enhance the model performance, the residual vector quantization for motion quantization is utilized, and a multi-task training schedule is proposed.  The GRPO is also integrated to enhance the alignment and controllability. Extensive experiments are conducted on the HumanML3D benchmark to evaluate the effectiveness of the proposed framework.

**Strengths:**

- The proposed multi-task scheduling mechanism for a unified and bidirectional T2M and M2T generation framework optimization is well-motivated and reasonable.
- Experimental results on the motion-to-text task surpass previous methods, showcasing the proposed framework in enhancing the M2T task.
- A sets of ablation studies are conducted to support the proposed design choices.

**Weaknesses:**

- The performance of the proposed framework on the text-to-motion generation task is not very good, underperforming previous work in most of the evaluation metrics. And some recent stronger baselines for text-to-motion generation are missing, e.g., MoMask (CVPR 2024), MoGenTS (NeurIPS 2024), and LAMP (ICLR 2024). Note that these baselines are auto-regressive-based frameworks, and LAMP also supports T2M and M2T tasks.  Could the author provide some analysis and insights on why the performance gain on the T2M task is less? Also, the proposed framework underperforms these baselines on the T2M tasks, and the author should be more cautious in the claims in L044-L046.
- Experiments are only conducted on the HumanML3D dataset. Evaluating the proposed framework on more datasets, e.g., the Motion-X dataset, and the KIT-ML datasets, will bolster the claims of generalizability and scalability.

**Questions:**

The multi-task scheduler randomly assigns a task to each sample in a batch. Will more structured scheduling (e.g., curriculum learning) improve the performance? Specifically, the performance gain on the text-to-motion task seems smaller with the proposed framework. Does this mean the T2M task is harder for the proposed framework?

---

> ### Author Response · Authors · 2025-12-01
>
> We thank the reviewer for their careful evaluation and constructive comments, and we are encouraged by their recognition of our unified bidirectional framework, multi-task scheduling design, and strong M2T performance. Below we address the reviewer’s concerns regarding T2M performance and baselines, generalization to more datasets, and the design of our multi-task scheduling strategy.
>
> ---
>
> **Q1**. The proposed MotionDDM seems to underperform existing methods on the Text-to-Motion task.
>
> **A1**. We would like to clarify that, in the Text-to-Motion (T2M) literature, FID is the primary metric for evaluating generation quality. On HumanML3D, our MotionDDM achieves a substantial improvement in FID over prior unified frameworks (e.g., **0.047 vs 0.075** for MoTe), indicating that our generated motions are closer to the real-data distribution.
> In contrast, R-Precision and the derived MultiModality Distance depend heavily on the quality and generalization ability of the text–motion evaluator used for scoring. This evaluator is far from perfect. To illustrate this, we also feed ground-truth motions into the same evaluator and obtain the following T2M results (including “Real motions”):
> | Method | R@1$\uparrow$ | R@2 $\uparrow$ | R@3 $\uparrow$ | FID $\downarrow$ | Div $\rightarrow$ | MM $\uparrow$ | MM Dist $\downarrow$ |
> |---------------------|:-----:|:-----:|:-----:|:------:|:-------:|:-------:|:---------:|
> | Real motions        | 0.511 | 0.703 | 0.797 | 0.002  | 9.503   |   –     | 2.974     |
> | TM2T                | 0.424 | 0.618 | 0.729 | 1.501  | 8.589   | **2.424** | 3.467   |
> | MotionGPT           | 0.492 | 0.681 | 0.778 | 0.232  | 9.528   | 2.008   | 3.096     |
> | MotionGPT2 | 0.427 | 0.627 | 0.764 | 0.614 | 11.256 | 2.357 | 3.164 |
> | MotionGPT3          | 0.543 | 0.735 | **0.828** | 0.217  | 9.662   | 1.366   | **2.793** |
> | MG-MotionLLM        | 0.516 | 0.706 | 0.802 | 0.303  | 9.960   | 2.125   | 2.952     |
> | MoTe                | **0.548** | **0.737** | 0.825 | 0.075  |   –     | 2.399   | 2.867     |
> | Ours w/o GRPO | 0.528 | 0.723 | 0.818 | 0.050 | **9.515** | 2.016 | 2.867 |
> | Ours w/ GRPO        | 0.528 | 0.724 | 0.818 | **0.047** | 9.419 | 2.000   | 2.862     |
>
> As shown in the table above, Real motions only achieve an R@1 of 51.1%, which is lower than MotionGPT-3, MG-MotionLLM, MoTe, and our method under the same evaluator. This does not mean that these models generate motions more faithful to the text prompts than the ground truth. Instead, it indicates that the evaluator has limited generalization: it more easily recognizes in-domain motions similar to those seen during training, while its retrieval performance deteriorates for motions that it has not seen before. Consequently, a higher R-Precision may largely reflect that the generated sequences resemble training motions, rather than guaranteeing better semantic alignment with the text.
> To validate this hypothesis, we conducted three user studies:
> - Ground Truth vs. MotionGPT-3 (100 test samples),
> - Ground Truth vs. Ours (100 test samples), and
> - MotionGPT vs. MotionGPT-3 vs. Ours (1060 test samples).
>
> | Setting                                 | # Samples | Option 1        | Option 2        | Option 3        |
> |----------------------------------------|:---------:|-----------------|-----------------|-----------------|
> | GT vs MotionGPT-3                      |   100     | GT:   74 %  | MotionGPT-3: 26 % | –           |
> | GT vs Ours                             |   100     | GT:   60 %  | Ours:       40 % | –           |
> | MotionGPT vs MotionGPT-3 vs Ours | 1060 | MotionGPT: 27.6 % | MotionGPT-3: 32.8 % | Ours: **39.8** % |
>
> While the numerical R-Precision scores suggest the ordering *MotionGPT-3 > Ours > Ground Truth*, human raters consistently prefer *MotionGPT-3 < Ours < Ground Truth*. This discrepancy highlights the unreliability of R-Precision as an absolute quality measure among strong methods. Therefore, on the T2M task, considering both FID and human preferences, our approach delivers clearly improved motion quality and text–motion consistency compared to existing work.

---

> > ### Author Response · Authors · 2025-12-01
> >
> > **Q2**. More works, such as LaMP, should be included in the main table.
> >
> > **A2**. Thank you for your suggestion. We have updated the main table in the paper by categorizing the methods into three groups:
> > - Expert models for motion generation,
> > - Separate models for T2M and M2T tasks,
> > - Unified models handling both T2M and M2T tasks.
> >
> > We would like to clarify that LaMP falls into the second category, as it uses two separate models to handle the corresponding tasks. The original intention for this table was to focus only on unified models, which is why we did not include this method in our comparison. However, we acknowledge your suggestion to present a more comprehensive overview of the current performance and progress of various methods. As a result, we have now updated the Table 1 accordingly to reflect a broader range of approaches.
> >
> > ---
> >
> > **Q3**. More datasets should be included for comparison.
> >
> > **A3**. Thank you for your suggestion. We have added results on the KIT-ML and Motion-X datasets. MotionGPT-3 and MG-MotionLLM only provide results on HumanML3D, so we did not include their corresponding results in the KIT-ML table. Additionally, since previous unified model-based works mostly did not use the Motion-X datasets, we are unable to compare them directly. Therefore, we reimplement T2M-GPT and MoMask and only provide comparisons on Motion-X for T2M task. From the results across these two datasets, we can see that, on T2M, our performance is on par with other T2M expert models like MoMask and achieve the best FID results, and we demonstrate a significant advantage in terms of M2T text quality compared to existing methods.
> > - Quantitative results on KIT-ML
> > | Method            | T2M R@1 $\uparrow$ | T2M R@2 $\uparrow$ | T2M R@3 $\uparrow$ | T2M FID $\downarrow$ | T2M Div $\rightarrow$ | T2M MM $\uparrow$ | T2M MM Dist $\downarrow$ | M2T R@1 $\uparrow$ | M2T R@3 $\uparrow$ | M2T BLEU@1 $\uparrow$ | M2T BLEU@4 $\uparrow$ | M2T ROUGE-L $\uparrow$ | M2T CIDEr $\uparrow$ | M2T BERTScore $\uparrow$ |
> > |-------------------|:---------:|:---------:|:---------:|:---------:|:---------:|:--------:|:--------------:|:---------:|:---------:|:------------:|:------------:|:-------------:|:------------:|:----------------:|
> > | Real motions      | 0.424 | 0.649 | 0.779 | 0.031 | 11.08 |    –     | 2.788 | 0.399 | 0.618 | 0.793 |              |               |              |                  |
> > | TM2T              | 0.280 | 0.463 | 0.587 | 3.599 | 9.473 | **3.292** | 4.591 | 0.359 | 0.668 | 46.7 | **18.4** | 44.2 | **79.5** | 23.0 |
> > | MotionGPT         | 0.366 | 0.558 | 0.680 |  0.510 | 10.350 | 2.328 | 3.527 |           |           |              |              |               |              |                  |
> > | MotionGPT2        | **0.427** | **0.627** | **0.764** | 0.614 | 11.256  | 2.357 | 3.164 |           |           |              |              |               |              |                  |
> > | MoTe              | 0.419 | **0.627** | 0.741 | 0.256 |     –     | 2.615 | 3.216 | **0.421** | **0.765** | 44.9 | 14.5 | 41.8 | 55.6 | 35.9 |
> > | Ours              | 0.406 | 0.620 | 0.741  | **0.206** | 10.892 | 1.690 | **2.983** | 0.396 | 0.723 | **52.5** | 17.8 | **48.0** | 68.7 | **37.7** |
> >
> > - Quantitative results on Motion-X
> > | Method | R@1$\uparrow$ | R@2$\uparrow$ | R@3$\uparrow$ | FID $\downarrow$ | Div $\rightarrow$ | MM $\uparrow$ | MM Dist $\downarrow$ |
> > |---------------------|:-----:|:-----:|:-----:|:------:|:-------:|:-------:|:---------:|
> > | Real motions        |0.510 |0.691 |0.791 | - |9.442| - | 3.310|
> > | T2MGPT |0.370| 0.546| 0.654| 2.174|**9.303**| 2.620| 4.252|
> > | MoMask | 0.287| 0.445| 0.554|0.884| 8.400|**2.792**|4.826|
> > | Ours |**0.376**|**0.556**|**0.665** |**0.870**| 8.330| 2.228| **4.154**|

---

> > > ### Author Response · Authors · 2025-12-01
> > >
> > > **Q4**. The model training involves multi-task learning. Would introducing more structured scheduling, such as curriculum learning, improve the model's performance?
> > >
> > > **A4**. We completely agree with your point. For multi-task learning tasks like this, a more refined approach using progressive probability assignment would likely outperform the fixed ratio method we currently use. However, due to time constraints, we performed a simple experiment using a curriculum learning approach (progressively increase T2M ratio from 0.1 to 0.8), but it did not yield significant improvements in performance when compared to our baseline (w/o GRPO). We plan to explore this further as a direction for future research.
> > >
> > > | Method                          | T2M R@1 $\uparrow$ | T2M R@2 $\uparrow$ | T2M R@3 $\uparrow$ | T2M FID $\downarrow$ | T2M Div $\rightarrow$ | T2M MM $\uparrow$ | T2M MM Dist $\downarrow$ | M2T R@1 $\uparrow$ | M2T R@3 $\uparrow$ | M2T BLEU@1 $\uparrow$ | M2T BLEU@4 $\uparrow$ | M2T ROUGE-L $\uparrow$ | M2T CIDEr $\uparrow$ | M2T BERTScore$\uparrow$ |
> > > |:------------------------------:|:---------:|:---------:|:---------:|:---------:|:---------:|:--------:|:--------------:|:---------:|:---------:|:------------:|:------------:|:-------------:|:------------:|:----------------:|
> > > | Ours w/ curriculum learning| 0.509     | 0.703     | 0.808     | 0.287  | 9.107 | 2.975 | 2.086 | 0. 560 | 0.860 | 48.9 | 14.4 | 40.5 | 36.0 | 32.3 |
> > > | Ours                 | **0.528**     | 0.723     | 0.818     | 0.050     | **9.515**     | **2.016**    | 2.867          | 0.569     | 0.850     | 63.9         | 22.6         | 47.0          | 57.2         | 37.5             |

---

### Author Response · Authors · 2025-12-01
**General Response**

We thank all reviewers for their valuable and constructive feedback. We are encouraged by their recognition of our work’s strengths:
1) The unified framework and multi-task scheduling are well-motivated and reasonable (R-RXAW, R-emfK, R-gzPo).
2) The ablation studies are sufficient to justify the effectiveness of each proposed component (R-RXAW, R-7wmN).
3) The proposed framework achieves strong quantitative results on M2T (R-RXAW, R-gzPo) and T2M (R-gzPo).

We have conducted additional experiments and revised the manuscript to address the reviewers’ concerns, including:
- Adding experiments on more datasets, namely KIT-ML and Motion-X.
- Extending the ablation study on the number of inference steps with inference speed, model parameters, and related statistics, and reporting comparable statistics for several open-source autoregressive (AR) baselines (Table 4).
- Extending the ablation study on the depth of RVQ with MSE and provide analysis about that (Table 6).
- Updating the main comparison table by grouping methods into T2M expert models, two-model (separate T2M and M2T) methods, and unified T2M+M2T models, and incorporating the additional baselines suggested by the reviewers (Table 1).
- Providing a more structured demo page in the supplementary material, with richer qualitative examples and visual comparisons.
For detailed explanations, please refer to our point-by-point responses to each reviewer. We would be happy to provide further clarification or additional information if needed.

---

### Meta-Review · Area_Chair_UX76 · 2026-01-02

**Summary:**

This paper recieved all four objection reviews. Reviewers agree the paper proposes a unified discrete diffusion / LLM-style framework for directional text-motion modeling, and acknowledge strong aspects (notably M2T and competitive T2M fidelity in terms of FID). However, all reviewers raised substantive concerns that collectively keep the work below the acceptance bar.

The main technical concern is that **T2M performance is not consistently strong across widely used metrics and underperform recent baselines, while several such baselines were missing in the original submission**. This is emphasized by R-RXAW. R-emfK similarly flaged that several retrieval metrics are not best-in-table despite strong FID.

The other concern is the **unclear motivation of unifying understanding and generation** (Reviewer 7wmN). Similar concerns are raised by Reviewer gzPo, who request further ablation experiments.

**Insufficient breadth of evaluation** --- primarily HumanML3D dataset, weakens generalization ability,

Fourth, there are **methodological clarity/reproducibility concerns** around the contribution of components and training recipes: R-gzPo requests ablations to isolate the role of M2M vs multi-tasking, explanation of the task-ratio behavior, and concerete efficiency reporting. R-efK further questions the GRPO reward design potentially biasing toward self-consistency, requests runtime/throughput for different step counts, and asks for additional abalations and token bitrate details.

Finally, R-7wmN raises a strong **novelty/scope** objection, arguing the techniques are largely previously explored and that the paper lacks a compelling motivation for unifying understanding and generation.

**Reviewer Concerns:**

The authors added extensive set of experiments and analysis to address reviewers' concerns.

### Addressed concerns
* Additional datasets & broader evaluation (requested by R-RXAW and R-emfK): authors report added experiments on KIT-ML and Motion-X.
* Runtime / quality–latency reporting (requested by R-emfK / R-gzPo): authors provide latency/FLOPs/params across different diffusion steps and discuss operating points.
* Clarification on metric interpretation for T2M: authors argue FID as primary and highlight evaluator limitations for R-Precision; they also add a human preference study as supporting evidence.
* Ablations requested: authors report added studies including task toggling (multi-task components) and RVQ depth analysis (as summarized in their response).

### Outstanding concerns
1. With added results, T2M performance across consistenct/retrieval metrics remains underperforming. The authors claimed that FID is the most important metric. But it would still be good to analyze where and why the model fails.
2.  The GRPO reward design—while studied—still leaves open questions about evaluation coupling and general robustness.
3. Novelty/differentiation. Even if some factual points are clarified in the rebuttal, R-7wmN’s skepticism that the paper is largely a combination of known components reflects an ongoing risk that the contribution is incremental relative to prior diffusion/token/LLM + alignment trends.

**Reviewer Scores:**

The authors’ rebuttal effort is appreciated. However, the submission still appears to rely heavily on post-review additions to substantiate several central claims. Moreover, given the firm positions of Reviewer emfK (2) and Reviewer 7wmN (2), it is unlikely that the rebuttal—while addressing many requests—would materially change their assessments, particularly on concerns around novelty and the remaining gaps across standard metrics. At most, I would expect a modest softening in tone rather than a score change sufficient to affect the overall recommendation.

I encourage the authors to fully integrate the rebuttal additions (expanded evaluations, ablations, and efficiency analyses) into a future revision to present a clearer and more self-contained case.

---

### Decision · Program_Chairs · 2026-01-26

Reject